# The transcriptional landscape of a rewritten bacterial genome reveals control elements and genome design principles

Mariëlle J. F. M. van Kooten [1✉], Clio A. Scheidegger[1], Matthias Christen[1] & Beat Christen [1✉]

Sequence rewriting enables low-cost genome synthesis and the design of biological systems with orthogonal genetic codes. The error-free, robust rewriting of nucleotide sequences can be achieved with a complete annotation of gene regulatory elements. Here, we compare transcription in *Caulobacter crescentus* to transcription from plasmid-borne segments of the synthesized genome of *C. ethensis 2.0*. This rewritten derivative contains an extensive amount of supposedly neutral mutations, including 123'562 synonymous codon changes. The transcriptional landscape refines 60 promoter annotations, exposes 18 termination elements and links extensive transcription throughout the synthesized genome to the unintentional introduction of sigma factor binding motifs. We reveal translational regulation for 20 CDS and uncover an essential translational regulatory element for the expression of ribosomal protein RplS. The annotation of gene regulatory elements allowed us to formulate design principles that improve design schemes for synthesized DNA, en route to a bright future of iteration-free programming of biological systems.

[1] Institute of Molecular Systems Biology, Department of Biology, Eidgenössische Technische Hochschule Zürich, Zürich, Switzerland.
✉email: marielle.van.kooten@alumni.ethz.ch; beat.christen@imsb.biol.ethz.ch

W e can program biological systems with DNA that is based on native nucleotide sequences. To enable DNA synthesis and assembly and to provide room for the creation of orthogonal genetic codes, native DNA can be rewritten. The introduction of extensive changes to the nucleotide sequence goes hand in hand with leaving gene expression and the gene product untouched and is, in part, enabled by the redundancy in the genetic code: the introduction of synonymous mutations in coding sequences is commonly referred to as recoding. We can rewrite non-coding sequences and synonymously recode protein coding sequences only with a complete map of the sequence-based information space. This space consists of the collection of transcriptional and translational regulatory features and the collection of stochastic processes that underlie RNA degradation, collectively employed in a cell to control gene expression[1]. As synonymous codon replacement retains only the amino acid sequence, these additional layers of information are erased. When George Church and coworkers[2] embarked to recode 13 codons in 42 highly expressed essential genes in *Escherichia coli*, they had already mentioned to expect some difficulties[3]. Indeed, for them and others, certain triplets proved to be recalcitrant to synonymous exchange. Since, systematic recoding of bacterial genomes has led to the identification and evaluation of design constraints[4–7]. Jason Chin and coworkers[8] showed that the exchange of even a single triplet can lead to detrimental effects in terms of phenotypic output, exemplifying that we have not yet captured all constraints necessary to synthesize biological information. Error-free, robust rewriting of DNA demands additional principles for the design of synthesized DNA.

In previous work, we synthesized and assembled the rewritten bacterial genome of *Caulobacter ethensis* (*C. eth-2.0* thereafter)[9]. The genome of *C. eth-2.0* comprises 676 protein-coding and 54 non-coding genes of freshwater α-proteobacterium *Caulobacter crescentus* (*Caulobacter* thereafter) compiled into a 786 kb genome design[10]. These genes were extracted from the native sequence as biological blocks (Designed Change 1) (Fig. 1a)[11], defined chunks of DNA that contain one or several genes that belong together from a gene regulatory point of view. Here, we compared transcripts originating from the genome of *Caulobacter* and from the plasmid-borne, segmented genome of *C. eth-2.0*. As interspersed sequences were omitted, genes are present in the same order, but with different neighboring elements (Designed Change 2). The genome has been rewritten—for one, we substituted bases to optimize the nucleotide sequence for chemical synthesis and DNA assembly. Second, we seeded 123'562 synonymous codon changes in protein-coding sequences (CDS) (Designed Change 3), collectively recoding 56.1% of all codons. Taken together, the rewritten genome of *C. eth-2.0* incorporates the transcriptional and translational control elements that are annotated. Any feature that has not been annotated will have been erased upon sequence rewriting. For the present study, we reasoned that the in-depth comparison of the transcriptional landscape of the rewritten and native genomes would reveal gene regulatory features and, due to the scale of the analysis, allow us to formulate principles to improve design schemes that allow rewriting DNA molecules whilst preserving biological functionality.

Collectively, we compared 612 protein-coding and noncoding genes by RNA-Seq[12]. We improved the annotation of 60 promoter regions and deduced the presence of 18 transcription termination

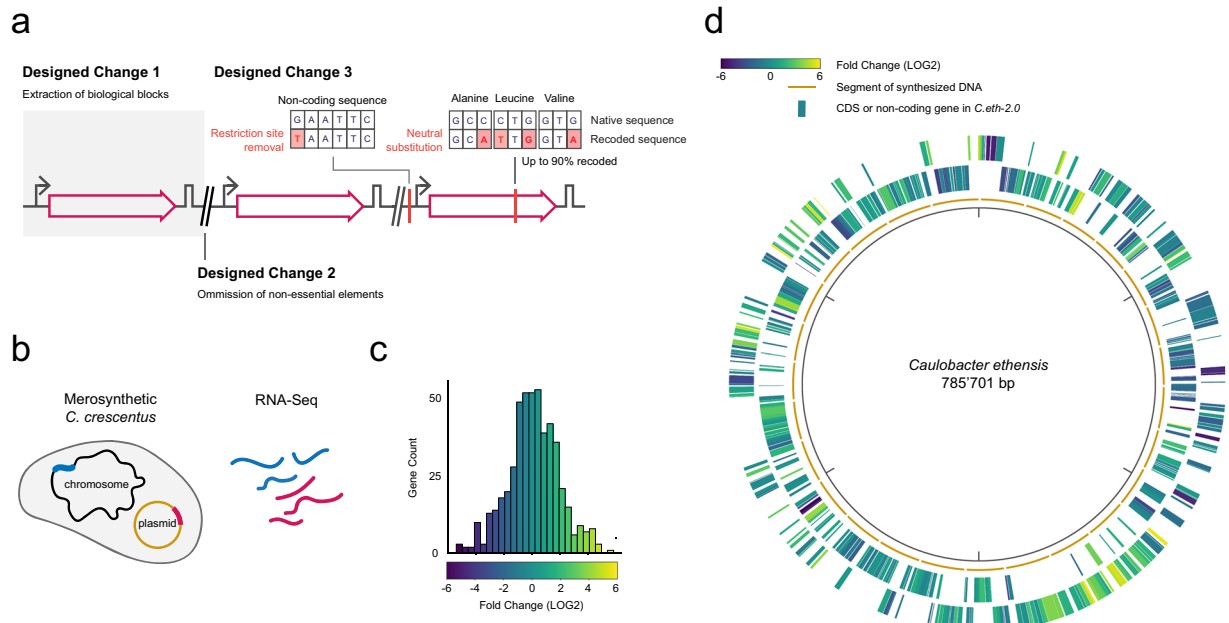

**Fig. 1 A genome-scale sampling of sequence-based information space to deduce global paradigms in genome biology that allow to improve sequence design and recoding schemes. a** Essential and high-fitness CDS as well as noncoding genes of *Caulobacter crescentus* have been isolated and rewritten: we removed obstructive features that hinder DNA synthesis and synonymously recoded up to 90% of a gene. Recoded DNA produces recoded RNA, but due to the redundancy in the genetic code, the protein sequence corresponds to the native sequence. **b** Merosynthetic *C. crescentus* contain a plasmid-based rewritten gene as well as the native copy of the gene on the chromosome and enable transcript comparison in a single experiment. The synonymous mutations introduced in the sequences of recoded genes allowed us to distinguish reads originating from the native and the rewritten genome. Blue: native gene and associated transcripts, magenta: rewritten gene and associated transcripts. **c** The LOG2 fold change in read count per gene derived through RNA-Seq for 506 measured genes and **d** for each recoded gene in the rewritten genome of *Caulobacter ethensis* when compared to native counterparts in the genome of *Caulobacter crescentus* for 506 measured genes present on segments of synthesized DNA that form the genome of *C. eth-2.0*. In the analysis of transcription level, genes were excluded based on the gene exclusion principles described in the Methods section. To focus on the expressed genes, the circular plot in panel (**d**) does not show genes that have been excluded. Outer rim: plus strand, inner rim: minus strand.

elements by transcript level comparison of native and rewritten genes. As a result of sequence rewriting, we unintentionally introduced 77 emergent transcription start sites (TSSs). These motifs form one of the main causes of transcriptome perturbation upon synonymous recoding. Last, we uncovered a conserved regulatory region internal to the coding sequence of tRNA methyltransferase *trmD* that facilitates the translation of *rplS*, coding for large subunit ribosomal protein L19P. Taken together, the transcriptional landscape of the rewritten genome of *C. eth-2.0* allowed us to uncover gene regulatory elements in the genome of *Caulobacter*. The genome-wide analysis led us to formulate general principles for the design of synthesized DNA, shedding light on bacterial genome biology in a broad context. As such, we contribute to a precise understanding of how to encode biological information into DNA, which will, in due time, enable a transition to the iteration-free programming of biological systems with synthesized information.

## Results

**Designer changes lead to the detection of transcriptional regulatory elements in the genome of *C. crescentus*.** To reveal gene regulatory elements in the genome of *Caulobacter*, we wanted to measure and compare the transcription of native, chromosome-based, and rewritten, plasmid-borne genes. The measurement necessitated the removal of ribosomal RNA (rRNA) as an abundant contaminant to enable NGS of RNA present even at relatively low abundance: each strain contained 4 Mb of chromosomal DNA counting for 3767 genes and 0.02 Mb of rewritten DNA, with, on average, 20 genes (Methods). A method for complementary oligonucleotide hybridization and RNase H-based degradation of RNA in RNA–DNA hybrids was developed in-house for the deenrichment of rRNA (Methods) based on published methods[13–15]. Removal of rRNA allowed for the acquisition of a higher number of strand-specific reads corresponding to mRNA (Supplementary Fig. 1, Supplementary Table 1), enabling us to capture transcription of plasmid-based rewritten genes (Methods). From the segment-wise RNA-seq measurements we reconstructed the transcriptome of the *C. eth-2.0* genome. Of the total amount of reads, $4.21\% \pm 2.26$ were mapped across the *C. eth-2.0* genome, showing that the rewritten genome is actively transcribed (Supplementary Data 1).

We had implemented three classes of Designer Changes in the rewritten genome. In the extraction of biological blocks (Designer Change 1), it is possible that we missed unannotated regulatory elements that are required for proper transcription. To assess whether plasmid-based rewritten CDS carry with them all necessary regulatory elements for transcription, we compared the transcript levels (the fold change (FC) in read count per gene, RCPG) between the transcripts of 506 rewritten genes and their native counterparts (Fig. 1c, Methods, Supplementary Data 2 Sheet 2). We corrected read counts for plasmid copy number as acquired through a linear derivation based on reported copy numbers for RK2-based plasmids[16] (Methods, Supplementary Data 1 Sheet 2). As a starting point for the comparison, we used a threshold value of an absolute LOG2 FC of 1 in the assessment of element inclusion. This value is based on the common metric utilized to report transcriptome differences under contrasting conditions. More than 60% (314) of genes showed $-1 \leq$ LOG2 FC $\leq 1$ in transcription level, suggesting that native regulatory elements have been included in the majority of rewritten genes. A reduction of the transcription level to LOG2 FC $\leq -2$ in 17% (86) of rewritten genes pointed to the loss of transcriptional regulatory elements (Supplementary Data 2 Sheet 2). An examination of the native RNA profiles led us to identify 18 sites controlling more than a single gene and 42 sites controlling a single gene,

amounting to a total of 60 absent gene regulatory elements. These un- or misannotated elements are either present in nonessential genes (11) (Designer Change 2), in recoded genes (16) or have been mutated upon synthesis optimization (27) (Designer Change 3) or are situated more than 200 bp upstream of the gene (6) and the sequence extraction is incomplete (Designer Change 1) (Supplementary Data 2 Sheet 3). As such, we propose to increase the bp range of or repair the promoter element for these genes (Design Principle 1) (Fig. 2 and Supplementary Table 4).

Rewritten genes that show a reduction in transcription level are expected to have lost regulatory elements. Similarly, we sought to reason why genes would show an increase in transcription level. Transcription from an upstream location in the DNA sequence was detected for 49 rewritten genes that show a LOG2 FC $\geq 0$. In 27 genes, we found the sense-oriented, upstream neighbor of the gene that showed transcripts originating from an upstream TSS to not be essential and thus be omitted (Designer Change 2) (Supplementary Fig. 5a, Supplementary Data 2 Sheet 4). We examined the native genetic context for these genes. We found sequences that terminate transcription of the upstream neighbor to have been included, erroneously and as a result of incomplete annotation, only in part in seven of these genes. For the remainder of genes, we screened the regions where terminators were to be expected in the native genome against WebGeSTer DB, a transcription terminator database[17]. We found four stem–loop terminators and seven Rho-dependent transcription termination elements in non-essential genes that had been omitted from the rewritten genome (Supplementary Table 2). These terminators are necessary for transcription termination of the respective upstream genes.

Operon organization and transcriptional control features protruding into neighboring genes can complicate genome design. For 18 genes, the extraction of biological blocks (Designer Change 1) and the exclusion of non-essential genes (Designer Change 2) led to the erroneous omission of transcription termination elements. Our findings highlight the importance and opportunity to reconsider the RNA-Seq data available for an increasing number of genomes to map or improve the accuracy of the annotation of elements that are difficult to predict based solely on the sequence. We use these cases to improve design schemes and achieve better transcript separation (Design Principle 2) (Fig. 2 and Supplementary Table 4).

**Sequence rewriting inadvertently introduces promoter elements that lead to aberrant transcription.** While 49 genes showed transcripts originating from upstream locations in the rewritten genome, we found missing terminators for 18 genes. To understand how to better insulate the 31 remaining genes, we considered an orthogonal analysis approach to inspect the same RNA-Seq data (Methods). The transcription curve of a gene (Fig. 3a) is defined by the read counts per base at the genomic location and can uncover a spectrum of phenomena, among which sequence or structure-specific endonucleolytic cleavage[18], and is mainly used to assess TSSs and splicing events[19,20]. Interestingly, the underlying cause for 17 of these genes was not the omission of transcription termination elements, but rather the unintentional introduction of a TSS in a rewritten intergenic region or within an upstream recoded gene (Designer Change 3) (Supplementary Data 2 Sheet 4).

To assess the extent to which the massive scale of rewriting inadvertently introduces TSSs, we screened the transcriptome of *C. eth-2.0* for emergent TSSs (Methods). We uncovered 77 emergent TSSs, of which 63 occurred in CDS and noncoding genes (17 sense, 46 antisense) (Supplementary Data S2 Sheet 8A). In the identified regions, rewriting increases the raw read count

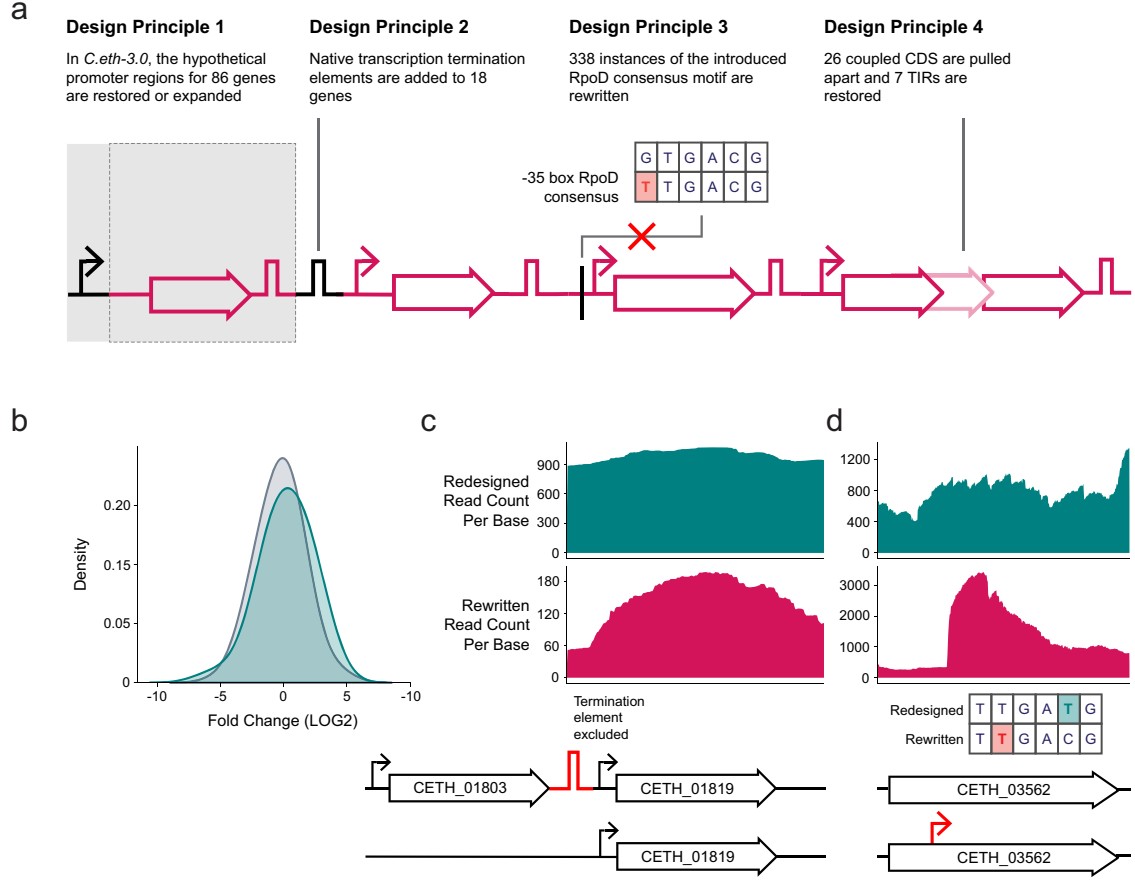

**Fig. 2 Principles for synthetic genome design, to be implemented in *C. eth-3.0*.** Based on observed changes in the transcription level and transcription curve, analysis of the nucleotide sequence, conservation, and structural analysis, and, for selected cases, experimental validation, we suggest to implement the principles depicted. We redesigned a subset of 82 CDS and noncoding genes according to these principles. **b** Expansion or repair of promoter regions increased the transcription level for 11 of 12 assessed genes and the distribution of the fold changes of all measured genes, depicted here as smoothed kernel density estimates, has shifted accordingly ($\mu$ −0.24, $\sigma$ 1.69 (gray) to $\mu$ 0.21, $\sigma$ 1.93 (teal)). **c** The placement of CETH_01803 in a different context to assess transcription termination shows read-through upon redesign into *hfq* (CETH_01819) **d**. Synonymous recoding of the promoter consensus motif in *atpA* (CETH_03562) removes the transcription start site that had been introduced unintentionally in *C. eth-2.0*. Teal trace: redesigned transcription curve derived from redesigned read count, magenta trace: rewritten transcription curve derived from rewritten read count.

per base (RCPB) by up to 22'466 from an initial 3'503 (Supplementary Data S2 Sheet 8A).

The emergence of TSSs upon sequence rewriting at scale necessitates the question how we can prevent it. In the set of emergent TSSs that we had identified, ample mutations had been introduced in the nucleotide sequences upstream of the TSS. An initial pattern search (Methods) revealed a motif with similarity to the −35 box and −10 box consensus for the principal sigma factor in *Caulobacter*, RpoD (5′-TTGaCgS-3′ and 5′-GCtA-NAWC-3′, respectively)[21] (Supplementary Fig. 3) to which 38 of 77 sequences contributed. Consensus motifs at both the −35 and −10 position form the landing sites that RpoD recognizes, after which transcription is initiated. We scanned 150 bp at each of the 77 TSSs for the −35 box RpoD consensus motif 5′-TTGNCNS-3′ and the −10 box consensus motif 5′-GCtA-NAWC-3′. The TSS strengths (Supplementary Data 2 Sheet 8A) of instances in which both motifs occurred and were present within 10–19 bp of each other (7 TSSs, Supplementary Fig. 4, Supplementary Data 2 Sheet 8B) were compared to the TSS strengths of instances in which either one of the motifs occurred or motifs were present at ranges that were deemed not biologically relevant (−35 box-like: 46 TSSs, Supplementary Data 2 Sheet 8C, −10 box-like: 23 TSSs, Supplementary Data 2 Sheet 8D). Although the small sample size and wide distribution

(mean and SD 2'064 ± 2'039, 4'018 ± 4'677, 2'845 ± 3'219, −35 and −10 box-like, −35 box-like, −10 box-like, respectively) did not allow a statistical comparison to show a significant difference, a visual inspection shows particularly strong TSSs to map to the occurrence of the −35 box motif (Supplementary Data 2 Sheet 8C), although the strongest instance (sense, *C. eth-2.0* genome coordinate 666'066) arguably comes with a −10 box-like instance that went unrecognized in our FIMO[22]-based search. In addition, a significant positive effect on TSS strength of the similarity of the motif to the consensus sequence 5′-TTGNCNS-3′ ($\rho = 0.33$, $p$ value = 0.02) was observed, suggesting that the reversal of single point mutations in this motif might allow to alleviate aberrant transcription in *C. eth-2.0*. We examined three cases in more detail, in which the −35 box consensus motif is present in the coding sequence in sense orientation and leads to transcription initiation (Fig. 4a). First, to confirm the strengths of the TSSs, we constructed a panel of transcriptional reporter fusions to *lacZ* comprising the regions that contained the hypothetical promoter elements in the rewritten sequence of CETH_03009, (corresponding to the observed instance at *C. eth-2.0* genome coordinate 588'240), CETH_03547 (see below), and *atpA* (CETH_03562, corresponding to the observed instance at *C. eth-2.0* genome coordinate 666'066) and the corresponding sequences at the native locations (Methods). Our assay confirmed that these

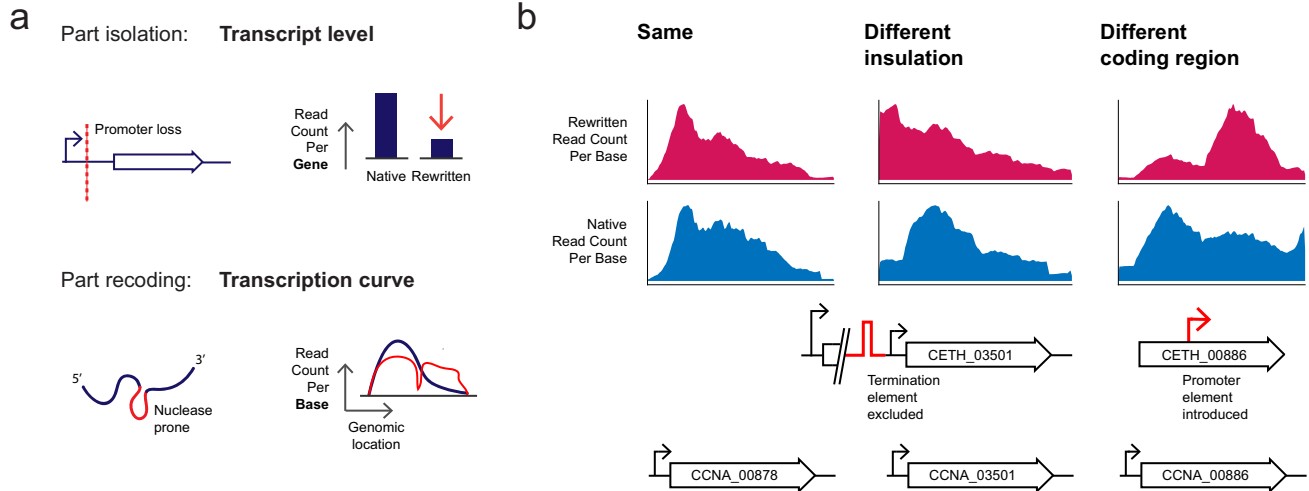

**Fig. 3 Comparison of transcription curves uncovers a distinct spectrum of phenomena and is a read-out complementary to transcription level. a** Perturbations that occur as a result of recoding of non-neutral sequences are quantifiable through analysis following RNA-Seq. Such perturbations are captured by the transcription curve as an analytical read-out complementary to the regular read-out of RNA-Seq, where transcript levels are compared. **b** The transcription curves of 601 genes were compared manually. Perfect transcription curves of a rewritten gene (CETH_00878 coding for ATP-dependent RNA helicase) with a recoded fraction of 0.27. CCNA_03501 encoding SUA5 translation factor-related protein, with a recoded fraction of 0.65, is preceded by CCNA_03502 in the native genome. In the rewritten genome, CETH_03501 is preceded by CETH_03510, as CCNA_03502 is not considered to be an essential gene. The altered context results in aberrant insulation. CETH_00886 (encoding aspartokinase, recoded fraction 0.22). Genes were manually labeled *same* or *different*, and *different* genes were sublabeled as displaying aberrant insulation, *insulation*, or perturbations that occur in the coding sequence, *coding region*. Magenta trace: rewritten transcription curve derived from rewritten read count, blue trace: native transcription curve derived from native read count.

sequences comprise strong promoters at 6'053 ± 137, 24'764 ± 602 and 14'210 ± 189 Miller Units, respectively, corresponding to a 30-, 83-, and 8-fold increase in $\beta$-galactosidase activity of the rewritten compared to the native sequences (Source Data). We then disrupted the -35 box consensus motif by a synonymous substitution of single nucleotides, for which we did not mutate back to the native sequences (Methods). These single point mutations reduced $\beta$-galactosidase activity 18- (345 ± 6 Miller Units), 18- (1'364 ± 21 Miller Units), and 6- (2'284 ± 82 Miller Units) fold, respectively (Fig. 4b, Source Data), confirming that the RpoD −35 box consensus motif 5′-TTGaCgS-3′[21] is one of the sequence motifs that contribute to the observed increase in RCPB.

Single point mutations may have considerable effects in a genome when sequences in that genome closely resemble promoter elements. In this situation, mutations allow for the emergence of novel gene products in amenable regions[23]. However, in addition to the absence of control and the implied cellular burden of extensive random transcription in synthesized DNA (Supplementary Fig. 2c), the necessity of a constraint in sequence design is strengthened by complementary observations in our data. CCNA_02635 encoding cell division protein FtsW encodes an isoform from a second ATG (ATG 2) at position M356 with a putative RpoD binding site 35 bp upstream of ATG 2[24]. Recoding has not abolished either feature in CETH_02635, and yet we no longer observe transcription from the second TSS (Supplementary Fig. 5b). *Cis*-suppression is a form of transcriptional interference, where an active upstream promoter suppresses transcription of its downstream neighbor[25,26], coined promoter occlusion by Adhya and Gottesman[27]. Interestingly, a RpoD consensus motif has been introduced in *ftsW* and might be responsible for decreased transcription from the native site present in the rewritten gene. To validate an instance of *cis*-suppression in *C. eth-2.0*, we measured transcription of CETH_03547 encoding peptidoglycan-specific endopeptidase LdpF in two contexts. CETH_03547 was measured both contained on single-segment

34 and in tandem with segment 33 (Supplementary Fig. 5c). In the former context, CETH_03547 is situated at the plasmid backbone-segment junction (Supplementary Data 2 Sheet 6). The backbone contains a promoter that fires into CETH_03547 and transcription was not observed from the introduced consensus motif. Next, transcription of CETH_03547 was measured when segment 34 was placed downstream of segment 33. In the absence of the plasmid-based promoter, the RpoD consensus motif served as a TSS. Our findings confirm *cis*-suppression to be possible even for strong promoter motifs that include the RpoD consensus motif. Finally, the faithful expression of rewritten genes can be affected by changes in RNA stability. *Cis*-encoded RNA, or antisense RNA originating from a promoter on the opposite strand is thought to be able to interfere with sense transcripts through the creation of double-stranded substrates that may form a target for endonucleases RNase III and E[28,29]. The results of either of these mechanisms would lead to mRNA read profiles as seen for *divJ* (CETH_01116) and CETH_03007 coding for L-aspartate oxidase (Fig. 4c, d).

Based on the ample generation of promoters upon synonymous codon replacement and the interference of such promoters with gene expression, we propose to put in place a coarse engineering constraint for sequence design by which we circumvent the introduction of −35 box promoter element 5′-TTGaCgS-3′ (Design Principle 3) (Fig. 2 and Supplementary Table 4). Future work that maps alternative and additional regulatory elements for transcription initiation at high resolution will refine this constraint to further decrease unintentional transcription and improve the robustness of designed sequences.

**Orthogonal transcription and gene functionality analyses of rewritten genes point to essential translational control elements.** Sequence rewriting can erase gene regulatory features that are not annotated. Information in protein-coding sequences, for instance, may encompass elements that regulate translation and

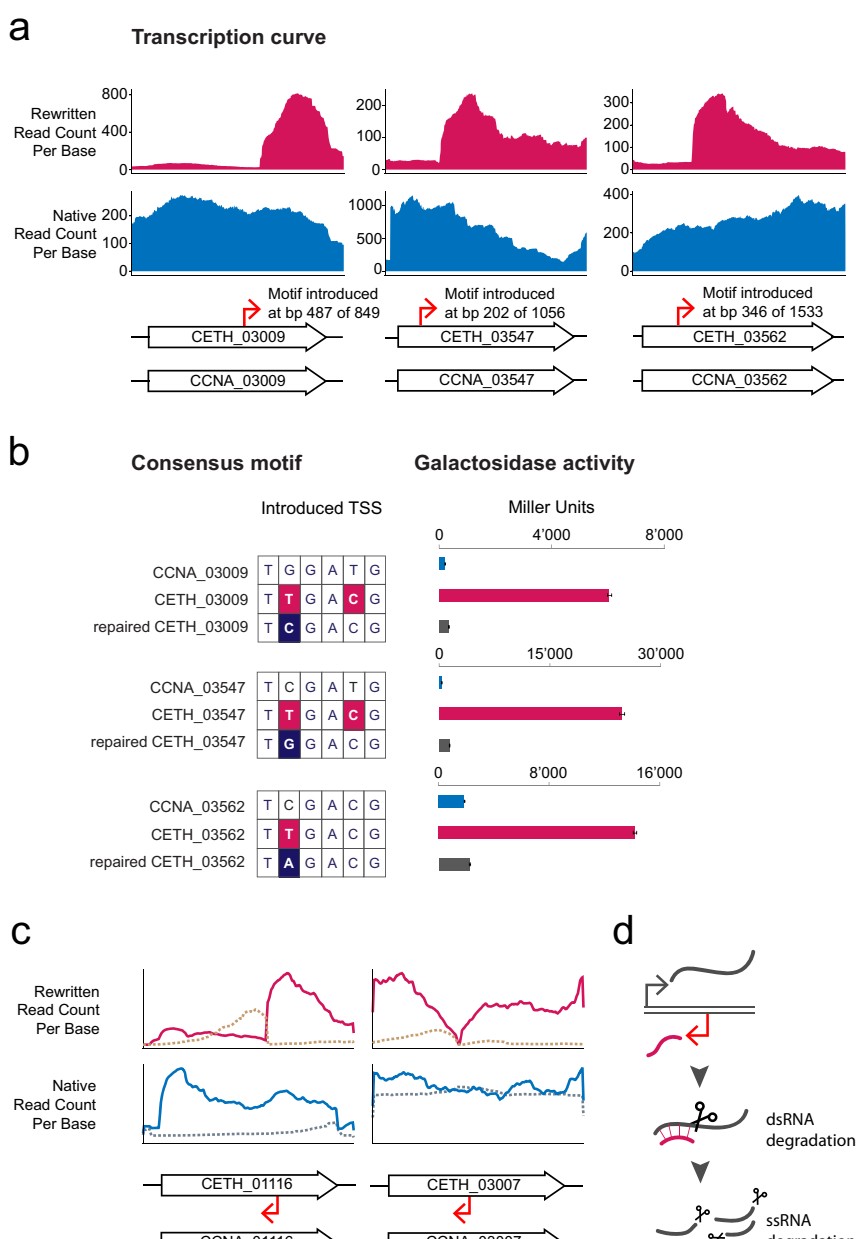

**Fig. 4 The emergence of novel transcription start sites. a** Emerging TSSs in intergenic regions or within upstream genes point to a −35 box consensus motif for the principal sigma factor in *Caulobacter crescentus*, RpoD. The promoter symbol and bp position mark the position of the consensus motif at −35 bp of the emergent TSS. The *x*- and *y*-axis have been adjusted to scale each set and to highlight the difference between rewritten and native transcription curves, respectively. CCNA_03009 is 849 bp, CCNA_03547 is 1056 bp, and *atpA* (CCNA_03562) is 1533 bp. Magenta trace: rewritten read count, blue trace: native read count. **b** Heavy firing can be significantly diminished by synonymous replacement of a single nucleotide within the consensus motif. Data are presented as the mean of *n* = 3 measurements ± the SD. Source data corresponding to individual data points is provided as a Source Data file. **c** Antisense interference in CETH_01116 (*divJ*) and CETH_03007. Dashed line: antisense rewritten (top) or native (bottom) read count. **d** The mechanism by which an emerging antisense TSS can lead to the depicted transcription curves of CETH_03007 by means of the creation of double-stranded substrates.

affect RNA degradation. To uncover such elements, we assessed the biological consequences of synonymous recoding through a comparison of the transcription curves of 601 CDS and noncoding genes. We labeled the transcription curves "same" (348) or "different" (253) (Methods, Supplementary Data 2 Sheet 5). Differences were observed to occur primarily due to poor insulation, a result of the absence of a terminator or the emergence of a novel transcription start site, causing read-through into neighboring genes. As such, genes with different transcription curves can be further divided into two categories: genes with differences in insulation (150) and a second category, in which we observed differences to occur mainly in the coding region (103) (Fig. 3b). Taking the amount of nucleotide substitutions into consideration—we introduced 123'562 synonymous codon changes alone—we observe a striking similarity (58%) in the transcription curves of rewritten genes and their native counterparts (Supplementary Data 2 Sheet 5), pointing to the absence of unannotated sequence-based information in the majority of native essential and high-fitness genes.

**Table 1 Comparison of the complementary measures functionality as derived through Tncite and transcription curve as derived through RNA-Seq as methods to pinpoint errors in genome design.**

| | | Transcription curve | | | |
|---|---|---|---|---|---|
| | | Same | Different insulation | Different coding region | |
| Functionality | Functional | 224 | 106 | 60 | 390 (83%) |
| | Faulty | 42 | 24 | 15 | 81 (17%) |
| | | 266 (56%) | 131 (28%) | 74 (16%) | 471 |

In previous work, we measured if a recoded, rewritten gene would be able to functionally replace the native gene or is faulty[9]. The results for 471 genes are contained in the rows "Functional" and "Faulty". The columns refer to the counts for these same genes with respect to the comparison of transcription curves between native and rewritten genes as observed after measurements described in this work.

In previous work, we measured if a recoded, rewritten gene can functionally replace the native gene or is faulty[9]. We reasoned that a recoded, protein-coding gene that shows an identical transcription curve but is unable to functionally replace the native gene will contain embedded post-transcriptional regulatory elements that we removed upon recoding (Designer Change 3). To probe this hypothesis, we integrated orthogonal measurements to annotate post-transcriptional control elements (Table 1). Of 471 rewritten genes for which transcription had been measured and that had been included in functionality measurements, 81 (17%) are faulty. Of these genes, 42 faulty genes displayed similar transcription curves to native genes (Table 1). To identify possible rewritten nucleotides that could hinder translation, we examined the translation initiation region (TIR) as the principal site of translational regulation in more detail (Methods). In 20 CDS, the TIR had been altered as a result of, most prominently, a recoded and coupled upstream gene or the introduction of mutations to enable DNA synthesis and genome assembly. With the changes between the rewritten and native sequences as a basis, we used in silico structural analysis, the presence of recoded Shine Dalgarno-like motifs, and conservation analysis to predict regions with control elements that affect translation (Methods, Supplementary Table 3). The analysis reveals such regions of putative regulation to occur predominantly in coupled genes. As such, in agreement with previous reports[2], we add essential constraints to the recoding of same-strand overlapping genes (Design Principle 4) (Fig. 2, Supplementary Table 4). We suggest to pull these genes apart, in effect duplicating the TIR and containing recoding to the upstream portion, preserving the putative control elements. Taken together, orthogonal transcriptome and gene functionality measurements allow to uncover putative control elements that function at the post-transcriptional level.

**Translation of large subunit ribosomal protein L19P, RplS is facilitated by a putative conserved mRNA hairpin in the coding region of tRNA methyltransferase TrmD.** Among the genes for which we predicted that unannotated, essential translational control elements are present in the rewritten TIR (Supplementary Table 3), we found the 4 ribosomal genes *rplS* (CCNA_00197), *rpsM* (CCNA_01328), *rpsK* (CCNA_01329), and *rpmF* (CCNA_01792). To examine if an unannotated translational regulatory element for *rplS* is necessary for faithful translation, we assessed the genetic configuration of this gene in detail. The ribosomal gene *rplS* encodes large subunit ribosomal protein L19P and appears to be translationally coupled with its ATG start codon overlapping the TGA stop codon of the preceding gene *trmD* (CCNA_00198), encoding tRNA (guanine-N(1)-)-methyltransferase TrmD. As a result, gene regulatory elements for the translation of *rplS* will lie in recoded *trmD*. To test for the presence of a translational regulatory element controlling translation

of *rplS* in *trmD*, we fused either the native or the recoded 3′ end of *trmD* to *lacZ* under the transcriptional control of the *trmD* promoter (Methods). Comparing the native and recoded reporter constructs, we observed a more than 6-fold reduction in β-galactosidase activity (WT 1'232 ± 22 Miller Units (MU), C. eth-2.0 202 ± 5 MU) confirming that the synonymous recoding of the 3′ end of *trmD* indeed removed an internal initiation signal necessary for faithful translation of *rplS* (Methods, Supplementary Fig. 6, Source Data). The 3′ end of *trmD* contains a putative Shine Dalgarno sequence that has been recoded to a less favorable motif in *C. eth-2.0*, making it a likely candidate to control translation of *rplS*. To assess if this element is responsible for the observed decrease in translational efficiency of CETH_00197, we introduced the native SD motif into the rewritten sequence. In this repaired construct, we measured β-galactosidase activity 2-fold higher (392 ± 4) (Supplementary Fig. 6, Source Data) than in the rewritten construct, not reaching a translation rate comparable to the native version, suggesting that additional unmapped sequences control translation.

To map the position of the additional putative translational regulatory element with nucleotide-level precision, we set out to establish the contribution of individual synonymous mutations in the 3′ region of *trmD* to the translation of *rplS* by scanning mutagenesis (Supplementary Fig. 7). First, we established the native and rewritten β-galactosidase activity in full-length, high-copy constructs that would allow to deduce even small effects (WT 11'359 ± 98, *C. eth-2.0* 226 ± 0 MU, Methods). Using translational LacZ reporter assays, we collectively probed the 15 individual synonymous codon substitutions within 100 bp upstream of the *rplS* start codon (Methods). The measurements uncovered a regulatory hot spot at Gly235 (2'343 ± 51 MU). Gly235 is situated in a conserved region predicted by structural analysis, SD recognition and conservation analysis, corroborating the predictive value of the approach (Methods, Source Data, Fig. 5), which had shown that this nucleotide may participate in the formation of an mRNA hairpin ($-13.24$ kcal mol$^{-1}$) that is conserved in *Caulobacter sp.* (Supplementary Fig. 7). Synonymous substitutions in CETH_00198 result in a less stable mRNA hairpin, although an effect of similar strength in recoded nucleotides downstream of Gly235 was not observed. To assess if translational control is a sequence- or structure-specific, additional synonymous mutations were introduced at strategic positions. First, we exchanged a predicted C-G bond in the native situation to a T-A bond (Methods). Where the single mutation of C to A dramatically decreases β-galactosidase activity (Gly235, 2'343 ± 51 MU), the double mutation decreases hairpin stability and thus β-galactosidase activity only slightly (9'606 ± 297 MU) (Source Data, Fig. 5b, Supplementary Fig. 7). In addition, a C to G mutation in Pro239 in the loop does not alter the stability of the putative mRNA hairpin. Indeed, we measured β-galactosidase at 11'876 ± 122 MU, similar to WT levels. Taken together, the

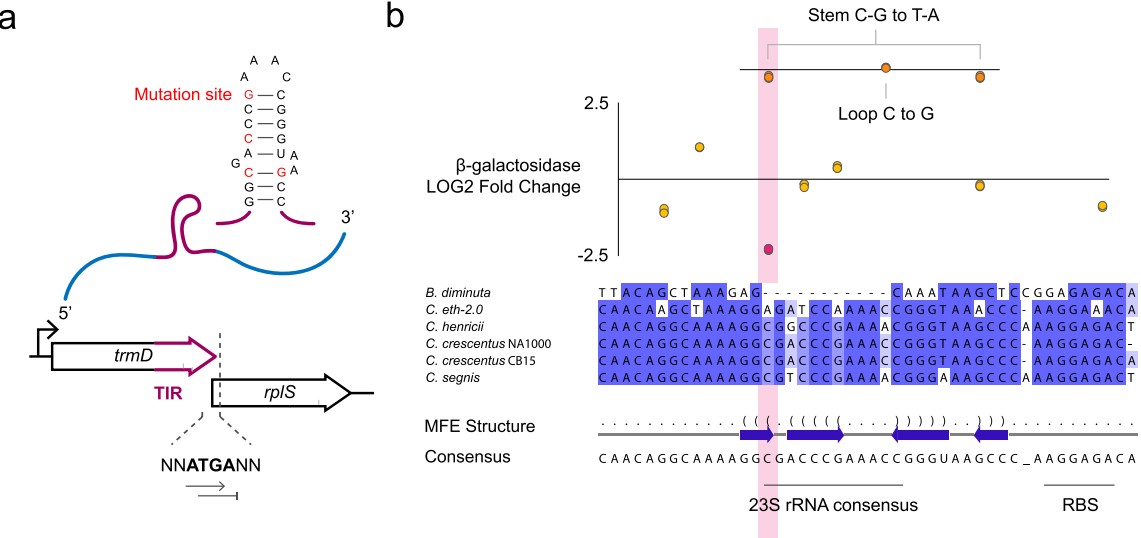

**Fig. 5 Translation of *rplS* is regulated through a conserved secondary structure in *trmD*. a** In silico structural analysis, the presence of Shine Dalgarno-like motifs and conservation analysis with the changes between the rewritten and native sequence as pointers converge to predict the presence of translational control elements in coupled CDS. The putative translation initiation regions of *B. diminuta* ATCC 11568 BDIM_RS13760, *C. sp. 'ethensis'* EUX21_00235, *C. henricii* CB4 AQ619_RS00715, *C. crescentus* NA1000 CCNA_00197, *C. crescentus* CB15 CC_0197 and *C. segnis* ATCC 21756 CSEG_RS1981 are depicted. **b** The 15 synonymous mutations, 7 of which are depicted here, introduced in the 100 bp upstream of the start codon of *rplS* were introduced in a translational fusion construct to *lacZ* on a single-nucleotide, base-by-base level. The nucleotide resolution reveals a regulatory hot spot (magenta) in the TIR of *rplS*. The mRNA species is predicted to contain a hairpin structure at −13.24 kcal mol$^{-1}$ that has a higher free energy upon recoding in CETH_00197. We have uncovered a 12 bp motif within the hairpin that is also and only present in 23S rRNA. The sequence that gives rise to a secondary structure is not hardcoded—synonymous recoding to sequences that allow base pair interaction or that are present in the loop is allowed (orange). Data are presented as one dot per independent measurement. Dots per sample amount to a total of $n = 3$. Dots corresponding to measured values for the stem C-G to T-A mutation are depicted at both mutations sites and amount to a total of $n = 3$. Source data are provided as a Source Data file.

$\beta$-galactosidase measurements of synonymous codon changes that abrogate hairpin formation together with the interchange-ability of nucleotides within the loop of the putative hairpin suggest that a structural RNA element may be necessary for efficient translation initiation of *rplS*. Captivating, however, is that the sequence folding into a hairpin contains 12 bp that are also—and only—found in the 23S rRNA in *Caulobacter* (Fig. 5b).

In summary, we suggest translation of ribosomal protein RplS to be facilitated through a conserved and essential RNA hairpin of 23 bp in the 3′ region of *trmD*. We expect that similar experiments for the 19 additional genes (Supplementary Table 3) will allow to converge from a genomic region to the single nucleotide level to elucidate essential translational control elements that were previously unknown.

**Implementation of the genome design principles improves the transcriptional landscape of synthesized DNA**. We contribute to the rules for the design of synthesized DNA by proposing 4 design principles that should bring the expression level and the transcription curve of rewritten genes closer to their native counterparts (Fig. 2 and Supplementary Table 4). To investigate if the transcriptional landscape of synthesized DNA indeed better resembles the native landscape upon implementation of the design principles, we redesigned a subset of 82 CDS and non-coding genes. For this subset, we expanded or repaired promoter regions (Design Principle 1), we placed genes in a different context to assess transcription termination (Design Principle 2) and we prevented the unintended introduction of promoters by electing against RpoD consensus motifs in rewriting (Design Principle 3). We assembled the synthesized DNA into 4 additional segments, introduced these segments into *Caulobacter*, creating 4 merosynthetic strains that each carry a subset of genes

in twofold, and measured transcription by RNA-Seq (Methods, Supplementary Data 3).

First, we assessed if transcriptional regulatory elements that were unintentionally omitted in the original design of the genome of *C. eth-2.0* had now been included in the repaired design (Design Principle 1) by comparing the transcription level of 12 redesigned genes for which we elongated and repaired promoter regions, using their native counterparts as a template (Supplementary Data 3 Sheet 2). The transcription level increased for 11 genes and the distribution of the FC of all measured genes shifted accordingly ($\mu$ −0.24, $\sigma$ 1.69 to $\mu$ 0.21, $\sigma$ 1.93, Fig. 2b). In summary, we improve the transcription level of genes for which regulatory elements had been excluded.

In a robust synthetic genome, transcription takes place in annotated regions. As such, we wanted to acquire a value to indicate if we successfully diminished aberrant transcription originating from both read-through and emergent promoter elements. To compare constructs of matching lengths in the native, rewritten and redesigned situation, we took the ratio of the RCPB in coding sequences and their antisense counterparts. This ratio is 13.86, 0.81, and 3.26 for the native, rewritten and redesigned constructs, respectively, indicating that we guide transcription to annotated rather than to unannotated regions of synthesized DNA. We then examined the transcription curves in more detail. Transcription termination of four genes for which we predicted to have omitted Rho-dependent or -independent termination elements was probed by the addition of native termination elements or by repositioning the genes involved (Supplementary Table 2). We observed read-through into *hfq* (CETH_01819), which had been positioned downstream of CETH_01803 upon redesign (Fig. 2 c). In our initial design, CETH_01803 led to read-through in CETH_01809. In the native context, transcription is terminated in CCNA_01806, which has

been excluded in *C. eth-2.0*. Finally, *pdxA* (CETH_01758) and *gmk* (CETH_01753) had been positioned in a convergent orientation in *C. eth-2.0*. Read-through had been observed from both genes in the rewritten constructs. Upon redesign, we removed CETH_01756 from the convergent center. Interestingly, the convergent orientation and the close vicinity in the revised context appeared to affect *pdxA* transcripts (Supplementary Fig. 5d). Taken together, the comparison of the transcription curves of rewritten and redesigned genes confirmed our annotation of the assessed transcription termination elements (Supplementary Table 2) and reflects the importance of the gene context. Next, we examined CETH_03547 and CETH_03562 in which we had observed and confirmed the presence of an emergent TSS (Fig. 4). We could confirm to have successfully removed the TSS by synonymous recoding of the RpoD consensus motif (Fig. 2d). As expected, solely using the exact match to the consensus motif, we do not remove all instances of emerging TSSs and introduce new sites on a limited number of occasions. In summary, we improved the robustness of the synthetic genome by assigning where termination elements are necessary and by removing elements that caused emergent TSSs.

Based on β-galactosidase measurements (Supplementary Fig. 6), the *trmD* promoter is required for transcription of polycistronic mRNA encoding *trmD* and *rplS*. To confirm our findings in order to decouple *trmD* and *rplS*, the 78 native bp upstream of the start codon of *rplS* were included in a redesigned biological block. The transcription level of decoupled, redesigned *rplS* at LOG2 FC −5.22 (Supplementary Data 3 Sheet 2) falls short and is lower than the level of coupled rewritten *rplS* at LOG2 FC −1.19 (Supplementary Data 2 Sheet 2). This signals that we, as expected, did not incorporate the regulatory elements for transcription into the biological block, omission of which is the main risk of decoupling. This also applies to CCNA_00320 and CCNA_00321, encoding LSU ribosomal protein L27P and L21P, respectively. We included the native 70 bp upstream of CETH_00320, of which 40 bp encodes the 3′ end of CCNA_00321. We observe a LOG2 FC of −6.00 for CETH_00320 upon redesign (Supplementary Data 3 Sheet 2), where we had measured a rewritten LOG2 FC of 0.54 (Supplementary Data 2 Sheet 2). For coupled genes, the strength of the upstream promoter and translational regulation, if any, are relevant to achieve the correct level of expression (Design Principle 4). As such, to decouple genes, we propose to consider the inclusion of the native, upstream promoter and to omit synonymous recoding in the 3′ region of the upstream gene.

We investigated 4 genome design principles and could confirm these principles to hold true upon their implementation in 82 redesigned genes. Experiments that relate these findings to translational fidelity, as we have shown for *rplS*, and to gene functionality[9] will provide the insights necessary to further develop and expand these principles.

## Discussion

We compared the transcriptional landscapes of *C. crescentus* and *C. eth-2.0*, a rewritten derivative in which we retained only annotated control elements and the amino acid sequence of protein-coding genes. Of the *C. eth-2.0* genes, 60% are transcribed at an absolute LOG2 FC ≤ 1 and 58% show a match to the native transcription curve. Simple design rules for sequence rewriting together with a well-annotated genome that prevents rewriting of the control elements not captured in those rules formed the starting point for genome rewriting. The resultant similarities in both gene expression and transcription profiles tell us that our starting point, to a reasonable extent, allows for the reliable programming of a biological system. Interestingly, as altered transcription patterns do not necessarily lead to an inability of the

rewritten gene to functionally replace the native gene (Table 1), we expect the cell to be able to cope with altered amounts of RNA and altered RNA processing to a certain extent on the one hand. Importantly, on the other hand, we do not know how translation will add to keep protein levels within the natural range[30,31].

Detected dissimilarities at the transcriptional and translational level allowed us to improve the robustness and functionality of the rewritten genome through the addition of design principles to a growing collection[2,4–6,8,9]. The design principles that we formulated contribute to capturing bacterial genome organization and control elements that are essential to robust and error-free gene expression. The guidelines that we deduced are relevant for any bacterial genome. Although the exact motifs differ between prokaryotes, transcriptional control is organized similarly across bacterial species: sigma factor binding sites control transcription initiation of genes or operons and transcription termination can take place in a Rho-independent or -dependent manner. As such, our design principles form the starting point that enables robust transcription initiation and termination in designed genomes. In conjunction, translation adheres to common principles - translation speed is dictated by, among others, local codon usage[32]. In addition, the Shine Dalgarno-like motifs that we circumvent introducing (Supplementary Notes) influence ribosome stalling[33–35], affecting RNA stability[36]. As such, the exclusion of known control elements in sequence rewriting is imperative to assess RNA stability as a result of nuclease activity, with the goal to make it a more controllable feature.

In the wake of previous reports, where overlapping genes in species such as *E. coli* had been considered[2], we examined translationally coupled genes in detail (Supplementary Table 3). We show orthogonal analyses to enable us to converge to nucleotide level resolution and uncover as of yet unknown sequence-based regulatory features for the expression of ribosomal gene *rplS* of *C. crescentus*, embedded within the 3′ region of the coding sequence of *trmD*. A possible model is the interaction of RplS with the 12 bp motif found in the 23S rRNA and the 3′ region of *trmD*. Translation of *rplS* would incorporate a negative feedback loop in which RplS binds to the TIR of *rplS* mRNA, hairpin folding cannot occur, secondary structure-aided (re) localization of ribosomes to the translation start site occurs to a less extent, and translation decreases. We have yet to corroborate such speculation, however, this type of regulation is not uncommon in the regulation of expression of genes encoding ribosomal proteins, such as L1 in *E. coli*[37–39].

Rewriting bacterial genomes or selected regions of such genomes enables their synthesis for biotechnological advances, and, as we show here, the discovery of regulatory information. The functionality testing of the components and scanning for regulatory information of these constructs is straightforward with the use of merosynthetic strains, in which plasmid-based portions of the rewritten DNA are cloned into the strain on which the components were based. Probing synthesized components at scale in vivo has been realized through the replacement of large sections of native DNA[4,5,7] while scanning DNA for regulatory information at high resolution can be achieved through RNA-Seq[12], ribosome profiling[40], or combinations of such methods[41] in native cells. The main advantage of the method we use is the method ease, in which we design, build and test rewritten sequences of any size within short periods of time without the necessity for whole genome synthesis, and flexibility in exchange of components as a result of the modular design. The latter results in a short turn-around time for the iteration and thus repair of design errors.

The design principles that we contribute here give genome engineers a higher likelihood of success when rewriting bacterial genomes. From a fundamental perspective, we improve the

annotation of transcriptional and translational control elements in *Caulobacter*, shedding light on genome organization in a broader perspective. Control elements and RNA degradation together govern gene expression. As such, we contribute to a high-resolution map of the nucleotide sequence-based information space. In due time, this map will enable programming of biological systems with synthesized information, allowing to explore possibilities that evolution has not yet touched upon.

## Methods

**Microbial growth conditions, media, and supplements**. Standard culturing conditions were used to grow microbial strains. *E. coli* was cultured in liquid or on solid Luria–Bertani medium at 37 °C while *C. crescentus* was cultured in liquid or on solid peptone-yeast extract (PYE) medium at 30 °C respectively. To resemble conditions used in Tn-seq[10] and Tncite[9], for RNA-Seq, *C. crescentus* was cultured in PYE supplemented with xylose. Cultures to be harvested for RNA-Seq were grown in baffled shake flasks or glass culture tubes at 250 rpm. Unless otherwise indicated, antibiotics and other supplements were used at the following concentrations in solid medium: *E. coli*: 50 µg ml⁻¹ kanamycin (km), 100 µg ml⁻¹ carbenicillin (cb), 30 µg ml⁻¹ chloramphenicol (cm); *C. crescentus*: 20 µg ml⁻¹ km, 50 µg ml⁻¹ cb, 2 µg ml⁻¹ cm, and in liquid medium: *E. coli*: 30 µg ml⁻¹ km, 100 µg ml⁻¹ cb, 20 µg ml⁻¹ cm; *C. crescentus*: 20 µg ml⁻¹ nalidixic acid (na), 5 µg ml⁻¹ km, 5 µg ml⁻¹ cb, 1 µg ml⁻¹ cm, 0.1% (w/v) xylose.

**Synthesis and assembly of redesigned DNA**. For the construction of 91'262 bp of DNA with 82 CDS and noncoding genes that were redesigned according to the design principles, DNA was ordered in 3- to 4-kb assembly blocks and segments were assembled into the pMR10Y shuttle vector by homologous gap repair in *Saccharomyces cerevisiae* following reported methods[9]. In brief, cells were collected from a mid-log phase *S. cerevisiae* culture by centrifugation, and washed. Salmon sperm DNA (single stranded from salmon testes, D7656, Sigma-Aldrich), linearized pMR10Y and column-purified assembly blocks were added. The pellet was resuspended in a transformation mixture consisting of PEG solution and lithium acetate in distilled water, followed by the addition of DMSO. Cells were incubated for 15 min at room temperature (RT) and for 15 min at 42 °C, harvested by centrifugation, resuspended, and plated.

**Construction of merosynthetic *C. crescentus***. Merosynthetic *Caulobacter* were used to enable transcript comparison by RNA-Seq[12] in a single experiment (Fig. 1b). The term *mero* derives from the merogenote state of an organism in which not all genes are present in multiple copies: merosynthetic *Caulobacter* contain a subset of plasmid-borne genes of the rewritten genome of *C. eth-2.0*. The synonymous mutations introduced in the sequences of recoded coding sequences (CDS) allowed us to distinguish reads originating from the native and the rewritten genome. In previous work, we had reported to have generated *E. coli* and *C. crescentus* that harbor 1–2 of 37 parts of the synthesized *C. eth-2.0* genome[9], referred to here as segments, in pMR10Y. Each strain carries 1 (31 strains) or 2 (4 strains) sequential segments of an approximate 20 kb of synthesized DNA. Each segment encodes 20 coding and non-coding sequences on average. For segments that are sequential in location within the rewritten genome, 4 plasmids were chosen containing a total of 8 segments that map to genomic locations 142'173 to 182'490 (segment 8 and 9), 161'246 to 202'617 (segment 9 and 10), 559'103 to 597'047 (segment 29 and 30) and 638'619 to 681'645 (segment 33 and 34). Of these 4 plasmids, 2 contain segment 9. The presence of segment 9 in 2 distinct constructs, 2 distinct strains was used as an internal quality control throughout the analyses. In this work, 4 segments comprising redesigned DNA were isolated from *S. cerevisiae* and transformed into *E. coli* DH10B by electroporation. All plasmids were conjugated from *E. coli* into *C. crescentus* NA1000 or NA1000 ΔrecA to generate strains of merosynthetic *C. crescentus*. Conjugation of pMR10Y to *C. crescentus* NA1000 ΔrecA led to the creation of a control strain.

**Sample preparation for RNA-Seq**. Merosynthetic *C. crescentus* were harvested in the exponential phase by centrifugation at 4000*g* for 10 min at 4 °C (Supplementary Data 1 Sheet 2). Strains were either pooled prior to harvest or following total RNA extraction. Cells were kept cool throughout sample preparation. A pellet corresponding to 10 ml of culture was redissolved in 1 ml of PBS and 5 ml of RNA-later® RNA Stabilization Solution (Ambion) was added. The mixture was incubated for 1 h at 4 °C. Following incubation, 7.5 ml PBS was added and the mixture was distributed into aliquots of 1.5 ml. Volumes were adjusted based on these ratios for smaller amounts of culture. After centrifugation, pellets were stored at −80 °C. Human recombinant lysozyme (Merck) was dissolved in Tris EDTA to 0.36 mg ml⁻¹. We added 100 µl to an aliquot of pelleted cells and incubated 5 min at RT. A volume of 400 µl of RLT (RNeasy Mini Kit, QIAGEN) supplemented with 2-mercaptoethanol (Merck) was mixed with the incubated cells, followed by the addition of 530 µl 70% EtOH. The procedure as described in the RNeasy Mini Kit with inclusion of DNase I treatment (RNase-Free DNase Kit, QIAGEN) was followed. RNA yield was quantified on a Nanodrop ND-1000 spectrometer (Thermo Fisher Scientific). Prior to content

analysis (Agilent 2100 Bioanalyzer, Agilent RNA 6000 Nano Kit, Agilent Technologies), secondary structure was diminished by incubation of samples for 2 min at 70 °C. To remove ribosomal RNA from the total RNA of *C. crescentus*, we followed two distinct procedures. As removal of ribosomal RNA by MICROBExpress™ Bacterial mRNA Enrichment Kit (Ambion) according to the manufacturer's instructions resulted in 88% of RNA mapping to ribosomal RNA, we used a second approach that is based on patented and published work[13,14]. We designed and ordered (IDT) 86 50-base DNA oligonucleotides covering the reverse complement of the entire length of 16S and 23S rRNA and followed the procedure as described in previous work[15]. We subsequently removed resultant DNA by DNase I treatment using an RNase-Free DNase Kit (QIAGEN) according to the manufacturer's instructions and purified using an RNeasy MinElute Cleanup Kit (QIAGEN). Depleted RNA was sent for library preparation and strand-specific, paired-end NGS on a HiSeq 4000 or NovaSeq 6000 S2 (Illumina) with a read length of 150 bp (GATC Services at Eurofins Genomics).

**Computational analysis of data acquired through RNA-Seq**. Strand-specific transcription of native genes that are chromosome-based and of rewritten, plasmid-based genes was measured in merosynthetic *C. crescentus*. The synonymous mutations introduced in the sequences of recoded genes allowed to distinguish reads originating from the native and the rewritten genome. Analyses were performed with bash, Python and, for DESeq, R. First, sequences were trimmed (Trimmomatic)[42], mapped (BWA-MEM)[43] and reads mapping to more than a single location were discarded. Second, general numerical characteristics of the processed data were acquired (SAMtools)[44] (Supplementary Data 1).

**Plasmid copy number normalization, pool correction, and between-sample normalization**. To normalize for plasmid copy number (PCN), general numerical characteristics were processed through a combination of two distinct methods. Normalization of counts derived from strains that had not been pooled prior to RNA-Seq was done according to method "Marczynski". The reported PCN of the RK2-based plasmid was plotted to plasmid size[16,45] and a linear equation PCN = −0.1017x + 7.3032 was fitted, where *x* is the plasmid size in kb (Supplementary Data 1 Sheet 2). The equation was used to derive the PCN of pMR10Y_ETH at 10.814 kb. Per sample, all reads that map to the plasmid-based *aph(3')-Ia* encoding aminoglycoside phosphotransferase, giving rise to kanamycin resistance (HTSeq)[46] were set to the control sample *C. crescentus* NA1000 pMR10, in other words, we corrected for the read count and assumed read count and resistance reads scale linearly. We then calculated which PCN corresponded to that amount of reads. Here, we assumed a linear fit. For method "Skyline", we acquired the RCPG (HTSeq). The read counts of rewritten and the (hypothetical) corresponding native segments were then forced on top of each other and a factor was retrieved that the rewritten counts should be corrected with to be able to do this. The score is an impression of the extent of deletions in a segment. Normalization of counts derived from strains that had been pooled prior to RNA-Seq was done as follows. A maximum of six strains was pooled either at harvest or following total RNA extraction. This meant that in each sample, the RNA that mapped to the chromosome was derived from multiple strains. For the acquisition of a correction factor that allowed to deduce the native chromosome count present in each individual strain, the measured absorbance was used for strains pooled prior to harvest or the total RNA quantity for strains where total RNA was extracted and subsequently pooled. In the analysis of part isolation, segments were excluded that contain a documented deletion or score under 1.0 in "Skyline". Of the total amount of 612 genes measured, we excluded 106 genes (see Methods, Gene exclusion principles). For the remainder of segments, we deployed "Marczynski" (Supplementary Fig. 2b), in which the linear equation was used to calculate the PCN that corresponded to the plasmid size present in each merosynthetic strain. Where relevant, to compare between samples, we let calculate a correction factor per sample (DESeq)[47] to correct the read counts for all genes in each sample.

**Acquisition and analysis of transcription curves**. Next, the RCPB was acquired (BEDTools)[48]. To compare transcription profiles, the RCPG was used to adjust the transcription profile: the FC was used to adapt the *y* value at each base location x of a transcription curve. This means that irrespective of the exclusion of regulatory features upstream of a gene that would lead to changes in RCPG, features that govern what happens during transcription of that gene can be compared. As an example, take a case where a promoter element has been rewritten and results in a lower RCPG, but features in the CDS are present as in the native situation. The area under the curve will have changed, but the curve is the same. Following segment- and gene-specific normalization, the resultant of the native and rewritten transcription curve of a gene was assessed (Python). For the assessment of transcription curves, first, genes were binned into three categories *same*, *different insulation*, and *different coding region* based on simple numerical features. As such, an increase in the area under the curve upstream of the annotated start would indicate aberrant insulation, an indication of e.g. the exclusion of transcription termination elements in the genome design. Along these lines, the introduction of promoter elements that lead to transcription start sites in coding regions, or the alteration of sequences that affect RNA degradation underlie changes in CDS and non-coding genes. To assess changes in the coding region of a gene, a gene-specific threshold of 0.5*mean of the RCPB for the native gene was set. Each base location above the threshold

added 1 to a total score. The total score was normalized for gene length, resulting in a number that represents the amount of a gene not corresponding to the native counterpart. If the resultant remained between a value of 0.5*median and 2.0*median, the deviation was discarded and called to be the resultant of a discrepancy in RCPG and RCPB. To further assess differences in coding regions, a comparison of the slope of a linear fit to the native and the rewritten gene was included. Differences in the slope would indicate altered degradation. Following categorization, transcription curves were plotted with the DNA Features Viewer Python library. A moving average was used to smooth the common, tiny fluctuations in curves to ease manual curve comparison. The gene categories were manually checked for genes that contained features that the algorithm did not accurately capture, for instance due to the presence of multiple features. In these cases, the gene was reassigned. Where obvious differences in coding region and differences in insulation were present, priority was given to the former.

**Analysis of off-target oligonucleotide hybridization in SDRNA.** To predict off-target oligonucleotide hybridization, the (up to) 50 bp oligonucleotides were mapped to the C. crescentus and C. eth-2.0 annotated transcriptome using BLASTN 2.11.0+[49] with default settings. To exclude location-specific degradation events arising from the rRNA depletion method, we compared the native mRNA profiles of the 12 examined CDS (Supplementary Data 2 Sheet 7) and the rewritten mRNA profiles of CETH_00899 derived by RNA-Seq of samples depleted with either MICROBExpress™ Bacterial mRNA Enrichment Kit (Ambion) or SDRNA. The absence of sequence similarity at the genome-scale and the absence of major discrepancies in the mRNA profiles of the 12 examined CDS led us to reject the hypothesis that the RNA profiles were technical artifacts of the depletion method.

**Gene exclusion principles.** The C. eth-2.0 genome comprises 768 CDS and ncRNA, tRNA, tmRNA, and rRNA species. Prior to transcript-level analysis, we sequentially applied gene exclusion principles. First, 2 genes where an unlabeled copy was present were discarded (CETH_R0045 and CETH_R0080, $768 - 4 = 764$). Next, genes that do not match in length ($764 - 23 = 741$), that have not been recoded ($741 - 52 = 689$), that have been denoted as present in regions that have been deleted in vivo ($689 - 64 = 625$)[9] and that are not fully contained on a single segment ($625 - 13 = 612$) were discarded. In addition to the denoted exclusions, for the analysis of isolated parts, genes ($612 - 106 = 506$) contained on segments that showed aberrant behavior upon PCN correction were excluded (Supplementary Fig. 2b). For the analysis of recoded parts, prior to transcription curve comparison, genes were discarded where the native and/or rewritten RCPG was below 16 ($612 - 11 = 601$) after depooling and application of plasmid copy number correction. The aberrant behavior upon PCN correction that leads to the exclusion of genes in transcription level comparison does not hinder transcription curve comparison.

**Exclusion of measurement artefacts of gene expression that trace back to the organization of genetic components and the customized carrier.** The RCPG of native genes was considered to assess if the distance from the origin of replication affects gene expression in C. crescentus. We did not observe a trend in RCPG of native genes and their location on the chromosome (Spearman correlation = −0.07, Supplementary Fig. 2a). Second, the topological state of DNA as a regulator of bacterial gene expression was considered[50,51]. Native, chromosome-based and rewritten, plasmid-based genes, when affected in a discriminate manner, would show a displacement in the mean from 0 in the FC distribution. Should genes on the chromosome be less accessible, for instance, we would expect the bulk of rewritten genes to be expressed at a consistently higher amount, displacing the mean to a more positive value. We do not observe a significant contribution of this phenomenon ($\mu$ 0.10, $\sigma$ 1.84) (Fig. 1c). Last, should feedback regulation govern expression upon introduction of additional copies of a gene, a non-normal FC distribution of native, chromosome-based, and rewritten, plasmid-based genes would be expected. Based on the distribution symmetry, such global trends are surprisingly absent in transcription. In summary, we did not observe major contributions of gene location, topological state of the DNA or copy number interference at the genome-scale when we considered the expression of native, chromosome-based genes and compared the expression of these genes to rewritten, plasmid-based genes.

**Identification of transcription start sites and sequence motifs.** An in-house Python script was employed to identify locations in C. eth-2.0 at which the sum of the raw RCPB measured across samples was under 4'000 with, in the next 100 bp, a 4-fold minimum increase with the raw RCPB over 1'000. Locations in the vicinity of annotated start sites and segment boundaries were excluded. The nucleotide sequences corresponding to the 150 bp upstream of the location at which the RCPB had increased to over the threshold FC and value were taken (Supplementary Data 2 Sheet 8A) to perform a pattern search. We employed MEME version 5.2.0[52] in classic mode while expecting zero or one occurrence of a contributing motif site per sequence, with the expectation of motifs being between 6 and 40 bp wide, and while searching the given strand only, with all other parameters set to default. To scan the nucleotide regions underlying the TSSs for the RpoD −35 and −10 box consensus motifs, we used FIMO[22] on two separate occasions with motif searches

for TTGACGS and GCTANAWC. We took into account matches with a p value below 0.01 and searched the given strand only. An in-house Python script was deployed to filter reported instances where multiple −35 or −10 box-like motifs were found in the same 150 bp, where only the instance with the highest FIMO score was retained. Next, −35 and −10 box instances within a 150 bp region that occurred within a stop-to-start distance of 10–19 bp[21] were called. Finally, for regions with both a −35 box and a −10 box at distances not in this range, we called the TSS to be under the influence of the box with the highest FIMO score.

**Statistical analysis of the effect of the distance from the origin of replication on gene expression.** To assess if the distance from the origin of replication affects gene expression in C. crescentus, the Spearman rank correlation coefficient between the RCPG and the location on the chromosome for 506 native genes was calculated (Spearman correlation = −0.07, Supplementary Data 2 Sheet 2).

**Statistical analysis of the effect of RpoD consensus motif resemblance on strength of emergent TSSs.** For the three sets (both instances, −35 box only, −10 box only), we compared the TSS strengths and did not find a significant difference (Mann–Whitney U). Calculation of a Pearson correlation coefficient of the FIMO score and the TSS strength suggests a significant positive correlation for −35 box instances ($\rho = 0.33$, p value = 0.02).

**Analysis of candidate TIRs to uncover control elements that enable post-transcriptional regulation.** Ten genes were excluded from further analysis due to, predominantly, mutations that had been introduced during the process of synthesis and assembly. As expected, such mutations would indeed not alter the transcription curve but could render the protein product not functional. For putative polycistronic genes, the 100 bp upstream of the start codon of the CDS was considered. For genes that did not appear to be transcriptionally coupled, the region up to the upstream neighbor in the native situation was assessed or, when this region exceeded 100 bp, the first 100 bp upstream of the start codon. Upon structural analysis of 100 bp regions upstream of the start codons of the corresponding CDS in RNAfold 2.4.11[53,54], stability of the first secondary structure encountered in 3′ to 5′ direction was calculated with RNAeval 2.4.11, in which the temperature was rescaled to 30 °C. SD-like sequences in these regions were manually assessed. The genomes of Maricaulis maris MCS10, Brevundimonas diminuta ATCC 11568, Caulobacter henricii CB4, C. segnis ATCC 21756, C. crescentus CB15, and C. crescentus NA1000 were derived via the NCBI Reference Sequence Database[55] and were subjected to conservation analysis of the nucleotide sequence with MUSCLE[56,57]. Jalview[58] was used to display alignments.

**Construction of strains for the validation of promoter elements that occur in coding regions.** Oligonucleotides (Microsynth/IDT) used in this study are listed in Supplementary Table 5. The nucleotide sequences of gBlocks (IDT) used in this study are listed in Supplementary Data 4. To show to have introduced a promoter element in the coding region of CETH_03009, CETH_03547, and CETH_03562, 99 base pair regions containing the hypothetical promoter element and an RBS followed by start codon ATG were fused to lacZ in pPR9TT[59]. Constructs were created through Polymerase Chain Assembly[60]. Included were the 99 base pair region at the corresponding location in the genome of C. crescentus as well as a copy of the rewritten region in which we had exchanged a single nucleotide. Inserts were fused to lacZ in pPR9TT through XhoI (NEB) and HindIII (NEB) restriction cloning.

**Construction of strains for the validation of post-transcriptional regulation in CCNA_00197.** Oligonucleotides (Microsynth/IDT) used in this study are listed in Supplementary Table 5. The nucleotide sequences of gBlocks (IDT) used in this study are listed in Supplementary Data 4. To show the promoter region of CCNA_00198 to give rise to a polycistronic mRNA, a 200 bp upstream region of CCNA_00197 up to and with the inclusion of the first 12 bp of CCNA_00197 was acquired through PCR[61]. To show the presence of a regulatory element in the TIR of CCNA_00197 and to probe the effect of the Shine Dalgarno sequence in the TIR of CCNA_00197, gBlocks (IDT) were designed in which the promoter region of CCNA_00198 was fused to the 100 bp upstream of CCNA_00197 with the inclusion of the first 12 bp of CCNA_00197, to CETH_00197 with the inclusion or to CETH_00197 with the natural SD site. Inserts were fused to lacZ in pPR9TT through XhoI and HindIII restriction cloning. To create high-copy plasmids, the above-mentioned 100 bp upstream of CCNA_00197 was extracted with the inclusion of the first 12 bp of CCNA_00197 and lacZ in pPR9TT through PCR. Fusion product pBVMCS-4-2 was constructed by ligation of the fragment and pBVMCS-4[62] following restriction with AscI (NEB) and AgeI (NEB), a high-copy plasmid that confers ampicillin and gentamicin resistance, carries an origin to enable conjugation and an MCS for translational fusion to lacZ. pBVMCS-4-2 was used to create a full-length translational fusion of the CCNA_00198 promoter, CCNA_00198, and the first 12 bp of CCNA_00197 and the CCNA_00198 promoter, CETH_00198 and the first 12 bp of CETH_00197 for base-level measurements and as a basis to introduce single mutations into the TIR of CCNA_00197. This was done by Gibson Assembly after restriction with KpnI (NEB) and SmaI

(NEB), with two fragments for either the native or rewritten construct, one of which was a gBlock (IDT), the other a fragment derived through (c)PCR. PCA of combinations of oligonucleotides yielded 17 fragments that incorporated the 15 mutations that were arbitrarily introduced into the TIR of CETH_00197 as well as double and single mutation-containing constructs that suggest the hairpin to be present as an essential secondary structure. The fragments were introduced into pBVMCS-4-2 through Gibson Assembly after digestion of pBVMCS-4-2 with KpnI and SmaI and with the inclusion of 2 fragments, one derived from PCR of pBVMCS-4-2 roughly corresponding to the CCNA_00198 promoter and CCNA_00198, and one roughly corresponding to the TIR for which ends were readjusted through PCR. For the latter, primer CXXX_00198-CXXX_00197 was reused.

**Construction of strains for the assessment of mRNA stability**. Oligonucleotides (Microsynth/IDT) used in this study are listed in Supplementary Table 5. The nucleotide sequences of gBlocks (IDT) used in this study are listed in Supplementary Data 4. To recapitulate RNA-Seq measurements and to probe whether removal of SD-like motifs in the coding region of CETH_00899 would allow stabilizing the mRNA, a panel of translational fusions to *lacZ* was created through Gibson Assembly of the PCR-derived CETH_00899 promoter and a PCR-derived fragment of CCNA_00899, or PCR-derived fragments of different lengths of CETH_00899 which included the CETH_00899 promoter, and PstI (NEB) and HindIII restricted pPR9TT. To recapitulate RNA-Seq measurements and to probe if wobble-paired rare codons are at the basis of mRNA instability, 2 translational fusions to *lacZ* were created by Gibson Assembly of gBlocks and PstI and HindIII-restricted pPR9TT. In each gBlock, either the sequence GGC GGC NNN GGC GGC or the sequence GGT GGG NNN GGT GGG was introduced downstream of the CETH_00899 promoter and the first 61 aa of CETH_03007, which was chosen for no particular reason other than to allow for translation to faithfully start. The sequence to be tested for effects on mRNA stability was positioned 14 aa upstream of *lacZ*. The β-galactosidase activities reported represent the average of two independent measurements derived from cultures in the exponential phase. Source data are provided as a Source Data file.

The gene expression in all constructs was created to be under the control of the CETH_00899 promoter. A comparison between the aforementioned and second panel of constructs showed gene expression in the second panel to be unexpectedly low. Upon detailed analysis, which included the consideration of ribosome profiling measurements[24], we realized that the translational start of CCNA_00899 is situated downstream of the annotated gene start at M22. The second panel of measurements does not have the CDS of CCNA_00899 at the basis, explaining the difference between the first and second panels of measurements.

**β-galactosidase reporter assay**. Reporter plasmids were conjugated into *C. crescentus* NA1000 ΔrecA cells and β-galactosidase-activity of the resulting translational reporter constructs was assayed in cells using standard ONPG based assays[63] at RT. Unless denoted otherwise, the β-galactosidase activities reported represent the average of at least three independent measurements derived from cultures in the exponential phase. OD600 at the time of harvest is denoted in the Source Data. Values are denoted as the mean and the standard deviation.

**Reporting summary**. Further information on research design is available in the Nature Research Reporting Summary linked to this article.

## Data availability
The RNA-Seq data have been submitted to the NCBI Sequence Read Archive under BioProject ID PRJNA695449. Supplementary Data 1 contains metadata, the data normalization procedure and the separation procedure for pooled strains. Supplementary Data 2 contains processed and analyzed data for rewritten genes. Supplementary Data 3 contains processed and analyzed data for redesigned genes. Supplementary Data 4 lists the gBlocks that have been designed and used in this work. Supplementary Data 5 lists strains used and created in this work. Source Data are provided with this paper. The Source Data contain the uncropped image displayed in Supplementary Fig. 1 b and the measurement values for the β-galactosidase measurements that have been performed. Supplemental data that support the findings of this study are available from the corresponding authors upon reasonable request. Source data are provided with this paper.

## Code availability
All software used in this study is denoted in the sections Results and Methods. The code to process and analyze RNA-Seq data derived from merosynthetic strains is available from M.v.K. upon reasonable request.

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

## Acknowledgements

The authors would like to thank J. Venetz, L. Del Medico, A. Wölfle, P. Schächle, Y. Bucher, D. Appert, F. Tschan, and C. Flores-Tinoco for generating mero-diploid reporter strains bearing segments of the *C. eth-2.0* genome, the Hall group and J. Hall for the use of laboratory equipment, GATC Services at Eurofins Genomics for the rapid processing of rRNA-depleted RNA, M. Okoniewski for the introduction to the Snakemake workflow and A. Typas and K. Weis for the assessment of the manuscript. This work received institutional support from the Swiss Federal Institute of Technology (ETH) Zurich, ETH research grant ETH-08 16-1 (to B.C.), and the Swiss National Science Foundation grant 31003A_166476, 310030_184664, and CRSII5_177164 (to B.C.).

## Author contributions

M.C. and B.C. conceived the research; M.v.K and B.C. designed experiments; M.C and B.C. performed genome redesign and assembly; M.v.K. prepared samples for the NGS of RNA; M.v.K. and C.S. performed validation experiments; M.v.K. conducted the experimental and computational analyses; M.v.K. and B.C. wrote the paper.

## Competing interests

The authors declare the following competing interests: M.C. and B.C. hold shares from Gigabases Switzerland AG. All other authors declare no competing interests.
