## [Peer Review File · Nature Communications]

Reviewers' Comments:

Reviewer #1:

Remarks to the Author:

In the manuscript "The transcriptional landscape of a rewritten bacterial genome exposes fundamental and engineered regulatory information" the authors evaluate the transcriptional function of a synthetic genome in a piecemeal fashion. Using plasmid-borne, 20-40kb synthetic segments introduced into an otherwise wild-type *Caulobacter crescentus* strain (a.k.a. merosynthetic *C. crescentus*), the authors transcriptionally profile re-designed gene sequences relative to the resident wild-type version. Their goal is to provide a more complete genome annotation and uncover biological properties of this organism that may be genomically encoded.

The manuscript needs to be revised for clarity throughout. Additionally, experimental testing of hypotheses generated is very much lacking in this manuscript.

At this time, I do not support publication of this manuscript as it needs major revisions.

General comments:

- The merosynthetic approach is an interesting idea and could be of broad interest. In particular, this approach could be leveraged across selected re-designed regions of genomes of interest and does not require total synthetic genome design and synthesis. This will make it far more attractive to a broad audience, compared to most projects related to whole genome synthesis. The authors should include a discussion of the limitations of the merosynthetic approach to genome annotation and compare it to other functional genomics approaches for genome annotation.
- The manuscript should be revised for clarity in nomenclature. The authors should clearly define the terms and then stick to those terms throughout (e.g. synthetic vs re-written vs re-coded seem to be used interchangeably and this is confusing in some places). The authors could also consider defining classes of designer changes that underlie experimental observations, and consistently using the nomenclature throughout. A few possibilities to consider: synonymous coding changes (base changes in coding sequences that do not alter AA sequence); intergenic sequence changes (e.g. for RE site elimination or base changes to make the sequence more synthesis friendly), formation of new junctions (e.g. between two essential genes during design when intervening non-essential genes were deleted), or between a synthetic segment and the plasmid backbone) etc.
- Use of the word "complementation" throughout the manuscript should be carefully thought through. The term suggests functional replacement of an otherwise mutant gene in the wild-type genome, which is not a feature of the experimental approach in this manuscript.

Specific comments:

Intro:

- Line 38 – reference 2019 PNAS synthetic genome paper.
- Fig 1A – is the blue gene supposed to be wild-type and the red one synthetic? Please clarify in the legend.
- Line 52-53 – it would be informative to include in this sentence the average number of coding sequences per 20-40kb synthetic segment.
- Line 53 – indicate how many strains comprise the set that were used for measurements for this manuscript and specify whether any segments were left out of the analysis and why.

Detection of transcriptional regulatory elements:

- Line 81-84 – this conclusion should be clarified, it is not clear what each of the points means in

this sentence mean.

- Fig S2a/b – there seem to be relatively large gaps in reads mapped that align with unannotated segments– were these segments not included in the analysis? There do not appear to be gaps in Figure 1(d), which is inconsistent with Fig S2. This should be explained somewhere in the manuscript.
- Line 85 – consider changing this sentence as follows: “whether plasmid-based synthetic coding sequences carry with them all necessary regulatory elements for transcription..”
- Line 87 – Why only 506 genes, when the introduction indicated 612 genes were included in the study and there are 676 in the entire genome?
- Line 88 – incorrect reference to Table S2?
- Were the boundaries of all of the 20-40kb segments designed to be intergenic? Is it possible that synthetic genes have been separated from their native regulatory elements by virtue of segment boundaries? If yes, this is an important point to highlight. If not, then what fraction of synthetic genes that were up or downregulated were at the leftmost or rightmost end of the segment? Is it possible that regulatory elements for this subset of synthetic genes may simply be encoded in the adjacent segment? This is a major drawback of the segmental aspect of the merosynthetic approach and should be indicated in the discussion.
- Line 95 – incorrect reference to Table S3?
- Line 98-100 – it isn’t clear what the relationship between synonymous re-coding and deletion of unannotated regulatory elements is in this sentence?
- Line 103-104 – there are two kinds of omission: (1) deleted during design of the synthetic genome and (2) exclusion from a particular 20kb synthetic segment. Do these need to be specified throughout the manuscript?
- Line 104-106 – a diagram would be helpful to explain this sentence.
- Line 117-121 – the authors should test a few of the hypotheses generated in the above paragraph.

An engineering constraint circumvents aberrant introduction of promoter elements:

- Line 124/Fig. 2Sc – it is not clear what the authors mean by “part isolation”, and more generally what this sentence means. The legend of Fig. 2Ss suggests this refers to evidence of transcription (RNAseq mapped reads) to regions not annotated as genes in the wild-type genome as a consequence of “arbitrary synonymous recoding”. I think the authors mean that they found that transcription starts and/or stops outside of annotated gene boundaries in some cases, but the text in the legend and in the main body should be clarified.
- Line 145-146 – If not already done, the authors should scan the designed genome sequence for other instances of TTGACG in relevant positions to determine how many times this sequence was introduced without affecting transcription. Providing this information would provide a much stronger argument for the inclusion of this constraint in future designs.
- General - Might be interesting to take a few of the segments and clone them into their vector backbone in both orientations and repeat experiment – this would test whether new junctions associated the vector backbone contribute to transcriptional changes in their data.

Sequence-based information in native CDS is captured in the major fraction of recoded CDS

- Line 154 – how was the classification carried out? Manually or otherwise?
- Line 159 – incorrect reference to Table S5?
- Line 167-168 – isn’t it just a hypothesis that the Shine Dalgarno-like motif underlies the observed shape change?
- Line 178-180 – wording should be changed to specify you have generated a series of hypotheses related to shape perturbations. The authors should scan the remainder of the designed genome for instances of Shine Dalgarno-like motifs or glycine rich regions to determine how often these elements do not coincide with shape changes.

Orthogonal measurements of the transcriptome and phenotype uncover essential control elements

- Line 196 – define “5’ coupled genes”.

Coding region of tRNA methyltransferase trmD facilitates translation of large subunit ribosomal protein L19P, rplS

- Line 219-221/Fig. S3 – there seems to be an inconsistency between the main text and figure legend with respect to systematic and common gene names. The main text specifies CCNA_00197/rplS while the supplementary figure legend says CCNA_00198/rplS, and vice versa for trmD.
- Fig. S3 – this figure needs to be made much clearer – it isn't obvious what hybrid synthetic-wild-type sequences were tested and how these impact the results.
- The last sentence of this paragraph needs clarification.

Translation of rplS is facilitated by a conserved secondary structure in trmD

- The authors aren't showing any of the beta-galactosidase results for this experiment so it is difficult to evaluate the results presented in this section.
- Line 257-268 – speculation should be moved into the discussion section.

Discussion

- Line 283 – another interpretation of this is that nearly half of the time the re-written gene doesn't express like wild-type, meaning that it is little more than a flip of the coin whether any given re-written gene will "work". I don't know that I would call this mastery of the rules of gene re-writing and this language should be toned down.

Reviewer #2:

Remarks to the Author:

The manuscript describes the analysis of transcriptomic data derived from synthetic gene sequences, recoded to only preserve known regulatory motifs, in a natural host. As non-annotated regulatory sequence was open to recoding in the synthetic gene design, the authors hypothesised that altered expression of synthetic genes could be used to uncover novel regulatory mechanisms. By comparing transcript profiles of synthetic genes and natural genes in a *Caulobacter crescentus* host, the authors uncover previously cryptic sequence features affecting gene transcript levels. The authors also identify some possible novel post-transcriptional regulatory features. The approach is a powerful way to uncover the sequence-based mechanisms affecting gene expression and to help define design constraints for future synthetic gene and genome projects. The precise design constraints being proposed could be more clearly stated and would be better supported by more experimental evidence testing them.

Comments

Several statements are made about this work enabling "iteration-free programming of biological systems with synthesized information" and that the authors "master the rules of robust programming of a biological system to a great extent" and similar. This is a major claim and I do not think that the manuscript demonstrates this at all. Whilst the methodologies to screen gene designs for transcriptional and translational perturbations show interesting trends and individual cases, there isn't a great deal of direct experimental evidence to back these hypotheses as complete general rules for sequence design. There is still a large gap between the findings of this manuscript and some of the claimed implications. As such, they should be substantially toned down.

The authors find several relationship trends between certain types of sequence and effects on transcription, and from there imply that resulting design constraints can be applied to future synthetic gene and genome design. The manuscript could be clearer on exactly what these constraints are and how they could be applied to the design process. Perhaps a table listing the design constraints emerging from this work would be useful to clarify this.

What proportion of the “faulty” genes have sequence changes that can be specifically identified by any new design constraints as potentially causing a problem with transcription and translation? Conversely do any of the constraints, if retroactively applied to the synthetic genes, predict that genes should show disruption of function when they do not?

The manuscript discusses the potential causes of changes in transcript coverage shape in the “Sequence-based information in native CDS is captured in the major fraction of recoded CDS” section, identifying common sequence features that may be causing these changes. It is claimed “we can pinpoint which mechanisms underlie changes based on sequence...” but I cannot see that this is substantiated by any strong experimental evidence. To make these claims, the authors should make alterations to the synthetic genes to comply with their new constraints to show that they can alleviate detrimental transcriptional changes.

Table 1 shows that of the genes classed as functional, ~43% show differences in coverage shape. Of the genes classed as faulty, ~48% show differences in coverage shape. Can this really be said to support coverage shape as a particularly powerful way to identify changes that affect functionality? Why is the frequency of difference in coverage shape so similar for functional genes?

In the manuscript, you say that faulty genes with the same coverage shape must have differences in translation causing the defect. One of these genes is investigated in depth, but did you find any indications in the other sequences as to what could be causing the problems?

The manuscript is quite difficult to read in places. I’d recommend a further proof reading. There are a few sentences like “Orthogonal coverage shape comparison and functionality measurements of the recoded bacterial genome of *C. eth* 2.0 show to uncover novel regulatory features, RNA-based control elements” that would benefit from reworking.

The terminology around synthetic genomes and *C. eth* 2.0 isn’t always accurately employed in this manuscript and I think that this could be confusing for the reader. The work in this manuscript takes a handful of synthetic genes at a time and characterises them in the context of a natural host cell. Even if you accept that a DNA sequence that doesn’t allow for cellular growth constitutes an existing synthetic organism, as claimed in the introduction, the characterisation those genes in small batches in a natural host context is hard pushed to be called a complete *C. eth* 2.0 transcriptome. The authors should be careful not to imply that a collection of genes separately characterised constitute a synthetic organism that has been characterised.

Line 317-318 – “These guidelines are relevant for any bacterial genome” – A fairly important point is only very briefly addressed here. *C. crescentus* is undoubtedly a good organism to perform this study in, given the tight control of gene copy number, but a little more discussion on how applicable these findings might be for other organisms would be good. Why can we assume that the guidelines derived are applicable to any bacterial genome? This is obviously a major factor in the wider impact of this study.

The synthetic genes are tested in sections rather than individually. Is it possible that being co-located on a DNA section with a particular transcription factor could have an effect on a gene’s expression? In such a situation, there would be increased levels of the transcription factor, leading potentially to changes in the background gene expression profile of the cell or possibly even direct effects on the expression of a gene being analysed. Is there any indication in your data of any such effects?

Line 191 – You state that you exclude 10 genes from further analysis due to mutations introduced in synthesis or assembly. Why weren’t they fixed to allow their inclusion?

Construction of merosynthetic *Caulobacter crescentus* - 7 of the 37 parts were tested in tandem with another part. Segment 9 was used as a control as it was present in two strains in combination

with 2 other different segments. Why didn't the authors include segment 9 alone as a control strain? Wouldn't this be a good way to determine any effects of having 2, rather than 1, segment in the cell? It might also give an indication as to whether other synthetic genes co-expressed in the same cell have an effect on other synthetic gene expression profiles.

Sample preparation for RNA-Seq - "Merosynthetic *C. crescentus* were harvested at an OD600 of 0.4 or at mid-log phase" is quite vague. Does this indicate that samples were consistently harvested at OD600 0.4, which is mid exponential, or were different samples harvested using different criteria? Several other parts in this section are not as clearly explained as they could be, e.g the description of the "SKYLINE" method and the process described in lines 394-396. This section is the describing the acquisition of the key data underpinning this study and so it's important that the methodology is clear.

Gene exclusion principles - I'm confused as to what the numbers in brackets represent. Are these gene numbers mutually exclusive or are they additive? It states elsewhere that 506 genes were included in the analysis but it's not clear how many in total were excluded.

Construction of strains for the validation of post-transcriptional regulation in CCNA_00197 - I may have missed it, but I can't see the sequences of the gBlocks used or the constructs generated in the supplementary information. These sequences should be included and their location referenced in this methods section.

β -galactosidase reporter assay - This section states that the error bars have been omitted in Figure 4b and 4c (and I'm presuming also in Figure S4). A version of these figures with error bars should also be included in the supplementary information for reference.

Figure 1 - The layout of this figure is quite confusing. The *Caulobacter ethensis* genome is not obviously part of subfigure 1c and it took me a while to notice that the colouring of the gene markings on the outer rings corresponded to the fold change in expression. It might be clearer to make the genome map its own subfigure with an additional fold change scale bar to reiterate this.

Figure 2a - I appreciate that the shape of coverage is the important thing being compared here, but the different scaling presumably being applied to the synthetic and natural data could be a bit misleading. I'd recommend either adding some sort of scale to the y-axis or adding a third plot for each gene showing an overlay of the two curves to clarify exactly what's being shown.

Figure 2b - It's not clear, either here or in the corresponding manuscript text, where these bases actually lie in the CDSs. It would be useful to indicate the positions of the bases.

Figure 2c - In the text, this is referenced in the sentence "Detailed examination of the sequence of *ftsW* (CCNA_02635), where the coverage shape shows differences in the coding region, reveals the interplay of such processes (Fig. 2 c)". So far as I can see, *ftsW* does not feature in this figure at all.

Figure 2d - This shows a proposed mechanism that I don't think is mentioned at all in the manuscript text. If a mechanism is being proposed, it should be discussed in the text.

Figure 4a - The expanded sequence showing the stop/start junction of the two CDSs is displayed in a way that readers might initially take to be the two DNA strands. I'd recommend tweaking the style of this.

Figure 4b/c - The splitting of these two sub-figures is confusing, I'd recommend just making the whole thing one sub-figure.

REVIEWER COMMENTS

Remarks of the authors to Reviewer #1 and #2:

We thank Reviewer #1 and Reviewer #2 for their extensive and thorough feedback – the time and effort that was spent is much appreciated and the comments have been instrumental in the revision of both manuscript text and content. Among others, we designed, synthesized and assembled 4 additional genome segments, each measuring more than 20 kb of DNA, according to our proposed design parameters. Measurements in merosynthetic *Caulobacter crescentus* led us to strengthen our hypotheses with respect to gene regulatory elements and genome design principles: our principles now include the expansion or repair of 60 promoter regions, addition of terminators to 18 genes, removal of 76 transcription start sites from the genome, appointment of 13 regions with decreased RNA stability, and the proposition to pull apart 26 coupled genes, with the presence of 7 additional translation initiation regions that remain to be validated. This information is summarized in Figure 2 and SI Table S4.

It came to our attention that the method we had developed in-house based on patented and published methods, Selective Depletion of ribosomal RNA (SDRNA), had been published just before the initial submission of our manuscript (Huang, Yiming, *et al.* "Scalable and cost-effective ribonuclease-based rRNA depletion for transcriptomics." *Nucleic Acids Research* 48.4 (2020): e20-e20). Several similar methods in additional recent publications comprised the community response to the discontinuation of Ribo-Zero, a popular commercial kit for the removal of rRNA. While the development of the method was a means to an end in the context of this manuscript, we chose to include short statements on the results in the Results and Methods section, and SI Fig. S1 and SI Table S1, to further strengthen these community developments.

Reviewer #1 (Remarks to the Author):

In the manuscript "The transcriptional landscape of a rewritten bacterial genome exposes fundamental and engineered regulatory information" the authors evaluate the transcriptional function of a synthetic genome in a piecemeal fashion. Using plasmid-borne, 20-40kb synthetic segments introduced into an otherwise wild-type *Caulobacter crescentus* strain (a.k.a. merosynthetic *C. crescentus*), the authors transcriptionally profile re-designed gene sequences relative to the resident wild-type version. Their goal is to provide a more complete genome annotation and uncover biological properties of this organism that may be genomically encoded.

The manuscript needs to be revised for clarity throughout. Additionally, experimental testing of hypotheses generated is very much lacking in this manuscript.

At this time, I do not support publication of this manuscript as it needs major revisions.

General comments:

- The merosynthetic approach is an interesting idea and could be of broad interest. In particular, this approach could be leveraged across selected re-designed regions of genomes of interest and does not require total synthetic genome design and synthesis. This will make it far more attractive to a broad audience, compared to most projects related to whole genome synthesis. The authors should include a discussion of the limitations of the merosynthetic approach to genome annotation and compare it to other functional genomics approaches for genome annotation.

We share the view of the reviewer in that the merosynthetic approach to test rewritten sequences is applicable to studies beyond this project. We added a paragraph discussing the possibilities and the limitations, along with alternative methods for genome annotation to the **Discussion (Line 453 - 464)**.

- The manuscript should be revised for clarity in nomenclature. The authors should clearly define the terms and then stick to those terms throughout (e.g. synthetic vs re-written vs re-coded seem to be used interchangeably and this is confusing in some places). The authors

could also consider defining classes of designer changes that underlie experimental observations, and consistently using the nomenclature throughout. A few possibilities to consider: synonymous coding changes (base changes in coding sequences that do not alter AA sequence); intergenic sequence changes (e.g. for RE site elimination or base changes to make the sequence more synthesis friendly), formation of new junctions (e.g. between two essential genes during design when intervening non-essential genes were deleted), or between a synthetic segment and the plasmid backbone) etc.

We performed an extensive revision of the manuscript. In this, we chose to refer to “native” and “rewritten” genes and use the term “synthesized” on specific occasions when we refer to a DNA molecule. To define the terms “rewritten” and “recoding”, we added the following sentences to the **Introduction**:

The genome has been rewritten - for one, we substituted bases to optimize the nucleotide sequence for chemical synthesis and DNA assembly. Second, we seeded 123'562 synonymous codon changes in protein coding sequences (CDS) (Designed Change 3).

We thank the reviewer for the excellent suggestion on defining classes of designer changes that underlie experimental observations, which we were happy to implement: we revised **Figure 1 a** to depict how Designer Change 1 (the extraction of biological blocks), 2 (the omission of non-essential elements, leading to the formation of new junctions) and 3 (rewriting, which includes synonymous recoding) lead from the genome of *C. crescentus* to the genome of *C. ethensis*. We point to these Designer Changes throughout the manuscript text. In addition, we now follow this consistent presentation to also point to the design principles that we derive from this study. We have added **Figure 2** and **SI Table S4** and refer to the Design Principles 1 - 5 throughout the text.

- Use of the word “complementation” throughout the manuscript should be carefully thought through. The term suggests functional replacement of an otherwise mutant gene in the wild-type genome, which is not a feature of the experimental approach in this manuscript.

We are aware that the term is used in the context pointed out by the reviewer. To avoid confusion, we exchanged the term for the phrase “functional replacement of the native gene” throughout the text.

Specific comments:

Intro:

- Line 38 – reference 2019 PNAS synthetic genome paper.

We adapted **Line 41-42** as follows:

*In previous work, we synthesized and assembled the rewritten bacterial genome of *Caulobacter ethensis* (*C. eth* 2.0 thereafter) (Venetz et al., 2019).*

- Fig 1A – is the blue gene supposed to be wild-type and the red one synthetic? Please clarify in the legend.

We revised the legend of **Figure 1**, which now states:

Blue: native coding sequence, Magenta: rewritten coding sequence.

- Line 52-53 – it would be informative to include in this sentence the average number of coding sequences per 20-40kb synthetic segment.

We added the following statement to the **Results** section, **Line 75-76**:

(..) each strain contained 4 Mb of chromosomal DNA counting for 3767 genes and 0.02 Mb of rewritten DNA, with, on average, 20 genes (Methods).

- Line 53 – indicate how many strains comprise the set that were used for measurements for this manuscript and specify whether any segments were left out of the analysis and why.

In the **Methods** section, we now indicate the total number of strains included in the original experimental setup as follows:

Each strain carries 1 (31 strains) or 2 (4 strains) sequential segments of an approximate 20 kilobase of synthesized DNA. Each segment encodes 20 coding and non-coding sequences on average.

In addition, we revised the **Methods** section “**Gene exclusion principles**”, in which we specify the total number of genes that were measured. We explain that complete segments or genes were excluded from further analysis based on plasmid abundance or predefined gene exclusion principles, in which we now specify exact numbers.

Detection of transcriptional regulatory elements:

- Line 81-84 – this conclusion should be clarified, it is not clear what each of the points means in this sentence mean.

We integrated Lines 81 - 84 into the **Methods** section “**Exclusion of measurement artefacts of gene expression that trace back to the organization of genetic components and the customized carrier**” to clarify why these results are essential to allow to continue the analysis.

- Fig S2a/b – there seem to be relatively large gaps in reads mapped that align with unannotated segments– were these segments not included in the analysis? There do not appear to be gaps in Figure 1(d), which is inconsistent with Fig S2. This should be explained somewhere in the manuscript.

Please be referred to **Methods** section “**Gene exclusion principles**”, which we have clarified (see the statement in response to the comment “Line 53 – (..)” in the above). We added the following statement to the legend of **Figure 1**:

*In the analysis of transcription level, genes were excluded based on the gene exclusion principles described in the **Methods** section. To focus on the expressed genes, the circular plot in panel d. does not show genes that have been excluded.*

- Line 85 – consider changing this sentence as follows: “whether plasmid-based synthetic coding sequences carry with them all necessary regulatory elements for transcription..”

We changed the sentence in **Line 87 - 88** to:

To assess whether plasmid-based rewritten CDS carry with them all necessary regulatory elements for transcription, (..)

- Line 87 – Why only 506 genes, when the introduction indicated 612 genes were included in the study and there are 676 in the entire genome?

Please be referred to **Methods** section “**Gene exclusion principles**”, which we have clarified (see the statement in response to the comment “Line 53 – (..)” in the above).

- Line 88 – incorrect reference to Table S2?

We acknowledge that the use of “Table” where an Excel sheet is meant might lead to confusion. We adapted all instances where we refer to a sheet to “Sheet”.

- Were the boundaries of all of the 20-40kb segments designed to be intergenic? Is it possible that synthetic genes have been separated from their native regulatory elements by virtue of segment boundaries? If yes, this is an important point to highlight. If not, then what fraction of synthetic genes that were up or downregulated were at the leftmost or rightmost end of the

segment? Is it possible that regulatory elements for this subset of synthetic genes may simply be encoded in the adjacent segment? This is a major drawback of the segmental aspect of the merosynthetic approach and should be indicated in the discussion.

The boundaries of all 20-40 kb segments were designed to overlap. In our computational analysis, we excluded 13 genes that map to these overlaps and are thus not contained on a single segment. It is noteworthy to mention that these genes have in fact been measured, however, in our automated computational setup, we exclude the overlap and thus the genes. We performed an analysis of genes at segment boundaries and added **SI Dataset 2 Sheet S6** to the Supplementary Information.

We acknowledge the drawback of the segmented measurement in that distant, unannotated regulatory elements will distort expression measurements. In fact, such elements will distort expression measurements regardless of the segmented approach: we included 100 bp upstream of the TSS in the biological block for genes without annotated promoter elements. Although bacterial sigma factor binding sites are situated in reasonable proximity of transcription start sites (TSSs), the action of bacterial enhancer-binding proteins (bEBPs) that bind remotely upstream of TSSs is required for the formation of the transcriptionally competent open complex of the σ^{54} holoenzyme, as reviewed in Rappas *et al.* (2007) and Bush and Dixon (2012). As such, in *Caulobacter*, multiple *cis*-acting flagellar transcription regulation (*flr*) elements are situated at an approximate 100 bp upstream of the TSSs of flagellar structural genes for which transcription is initiated by σ^{54} (Minnich and Newton, 1987, Mullin *et al.*, 1987, Mullin and Newton, 1989, Wu *et al.*, 1995).

- Line 95 – incorrect reference to Table S3?

We acknowledge that the use of “Table” where an Excel sheet is meant might lead to confusion. We adapted all instances where we refer to a sheet to “Sheet”.

- Line 98-100 – it isn't clear what the relationship between synonymous re-coding and deletion of unannotated regulatory elements is in this sentence?

The statement was removed upon revision.

- Line 103-104 – there are two kinds of omission: (1) deleted during design of the synthetic genome and (2) exclusion from a particular 20kb synthetic segment. Do these need to be specified throughout the manuscript?

There is one kind of omission. The omission occurs during genome design and can be split into three categories. (1) An element is not annotated. (2) The element annotation is incorrect. (1) An element is present in a non-essential gene and is not considered to be essential or related to fitness.

- Line 104-106 – a diagram would be helpful to explain this sentence.

We included **SI Figure S5 a** to display how omission of an upstream transcription termination element can lead to read-through in a rewritten gene.

- Line 117-121 – the authors should test a few of the hypotheses generated in the above paragraph.

In **Results section “Implementation of the genome design principles improves the transcriptional landscape of synthesized DNA”**, we report the validation of the omission of regulatory elements in 13/14 genes by elongation or repair of the promoter region and measurement of the transcription level upon redesign (**SI Dataset S3**). We confirm the presence of 4/4 termination elements by changing the genetic context of genes that we mentioned in our original measurements (**SI Table S2**).

An engineering constraint circumvents aberrant introduction of promoter elements:

- Line 124/Fig. 2Sc – it is not clear what the authors mean by “part isolation”, and more generally what this sentence means. The legend of Fig. 2Ss suggests this refers to evidence of transcription (RNAseq mapped reads) to regions not annotated as genes in the wild-type genome as a consequence of “arbitrary synonymous recoding”. I think the authors mean that they found that transcription starts and/or stops outside of annotated gene boundaries in some cases, but the text in the legend and in the main body should be clarified.

We observed that more transcripts map to intergenic regions of the synthetic genome than to the native genome. We suspected this to be a result of generally higher levels of transcripts mapping to the synthetic genome, however, we did not observe a correlation between the fold change of transcripts at annotated locations and at intergenic (unannotated) locations. We attributed this discrepancy to, as the reviewer points out, transcription starts and stops outside of annotated gene boundaries. We clarified the legend of **SI Figure S2** accordingly:

This discrepancy was attributed to the omission of transcription termination elements and introduction of transcription start sites.

- Line 145-146 – If not already done, the authors should scan the designed genome sequence for other instances of TTGACG in relevant positions to determine how many times this sequence was introduced without affecting transcription. Providing this information would provide a much stronger argument for the inclusion of this constraint in future designs.

We scanned the synthetic genome for instances of TTGACG. Excluding the 200 bp upstream of each gene and discarding instances that are within 200 bp of an upstream occurrence, there are 338 occurrences of “TTGACG” in the synthetic genome that are not in the native genome. Based on transcription profile assessment, 68 (20%) of these occurrences are thought to function as a TSS. As the reviewer is undoubtedly aware, transcription initiation necessitates the recognition of and binding to additional sigma factor binding motifs, which, for the σ^{70} family members, are centered around the -10 position in bacteria.

Based on the comment and the analysis in the above, we realized a thorough revision of this section was in place, resulting in **Results section “Sequence rewriting inadvertently introduces promoter elements that lead to aberrant transcription”** in which we chose an unbiased approach using both genetic sequence and transcription profile to assess introduced TSSs and underlying motifs.

- General - Might be interesting to take a few of the segments and clone them into their vector backbone in both orientations and repeat experiment – this would test whether new junctions associated the vector backbone contribute to transcriptional changes in their data.

Please be referred to the response to the reviewer comment on plasmid-segment boundaries and on the observations with respect to CETH_03547, as denoted in Supplementary Results section “Validation of an instance of *cis*-suppression in *C. eth 2.0*”.

Sequence-based information in native CDS is captured in the major fraction of recoded CDS

- Line 154 – how was the classification carried out? Manually or otherwise?

The classification was carried out manually. We attempted to classify genes based on numerical features. In this, we used methods that had been reported for the appointment of exon-intron junctions in eukaryotes (Bioconductor package *rnaSeqMap*) as well as hand-coded, simple features that would together capture shape. Despite providing a more objective method to classify genes and the potential of applying such classification to datasets of increasing size, we decided to not rely on numerical classification (1) to circumvent false label appointment for edge cases and (2) to not sidetrack the reader. We now include a thorough description of the labeling process in the **Methods section “Computational analysis of data acquired through RNA-Seq”, subsection “Acquisition and analysis of transcription coverage shapes”**.

- Line 159 – incorrect reference to Table S5?

We acknowledge that the use of “Table” where an Excel sheet is meant might lead to confusion. We adapted all instances where we refer to a sheet to “Sheet”.

- Line 167-168 – isn't it just a hypothesis that the Shine Dalgarno-like motif underlies the observed shape change?

The reviewer is correct. We realized this section needed to be placed in the broad context of research on RNA structure, ribosome dynamics and RNA degradation. The **Results section “Transcriptional interference and synonymous codon changes affect RNA stability”** has been added to better include the extensive multi-faceted research that has been done in this area, and to tone down speculation. Although we realize – and address – that follow-up experiments are necessary to shed light on the observed patterns, such measurements are out of the scope of this work. Nevertheless, we excluded technical artefacts resulting in the patterns that we observe (**Methods section “Analysis of off-target oligonucleotide hybridization in SDRNA”**), and feel our results are of considerable value to the research community. We are confident that this section will contribute to future work from multiple research groups, leading to a better understanding of RNA structure, ribosome dynamics and RNA degradation.

- Line 178-180 – wording should be changed to specify you have generated a series of hypotheses related to shape perturbations. The authors should scan the remainder of the designed genome for instances of Shine Dalgarno-like motifs or glycine rich regions to determine how often these elements do not coincide with shape changes.

Please be referred to the reaction to the previous comment.

Orthogonal measurements of the transcriptome and phenotype uncover essential control elements

- Line 196 – define “5' coupled genes”.

The statement was removed upon revision.

Coding region of tRNA methyltransferase trmD facilitates translation of large subunit ribosomal protein L19P, rplS

- Line 219-221/Fig. S3 – there seems to be an inconsistency between the main text and figure legend with respect to systematic and common gene names. The main text specifies CCNA_00197/rplS while the supplementary figure legend says CCNA_00198/rplS, and vice versa for trmD.

The reviewer is correct and we apologize for the confusion - we have adapted the instance in the legend of **SI Figure S6** (was: SI Figure S3), where CCNA_00197 is referred to as *trmD* and CCNA_00198 is referred to as *rplS*.

- Fig. S3 – this figure needs to be made much clearer – it isn't obvious what hybrid synthetic-wild-type sequences were tested and how these impact the results.

We adapted **SI Figure S6** (was: SI Figure S3) to depict how each construct is built. We adapted the **Results section “Translation of large subunit ribosomal protein L19P, RplS is facilitated by a putative conserved mRNA hairpin in the coding region of tRNA methyltransferase TrmD”** to clarify how each construct contributes to an understanding of why CCNA_00197 (*rplS*) was observed to be “faulty”.

- The last sentence of this paragraph needs clarification.

The statement was removed upon revision.

Translation of rplS is facilitated by a conserved secondary structure in trmD

- The authors aren't showing any of the beta-galactosidase results for this experiment so it is difficult to evaluate the results presented in this section.

All measurements of beta-galactosidase activity for the 17 strains that were generated for this experiment (WT and *C. eth 2.0* base level, 15 synonymous mutations that were included in the genome design, 2 additional mutation (pairs)) are included in the **Source Data**. In part, the data is presented in Figure 5 b (was: Figure 4 b and c). The data on the 15 mutations that had been included in *C. eth 2.0* are presented in SI Figure S7 (was: SI Figure S4).

- Line 257-268 – speculation should be moved into the discussion section.

Speculation with respect to the presence of a potential 23S rRNA consensus motif was moved to the **Discussion (Line 446-451)**.

Discussion

- Line 283 – another interpretation of this is that nearly half of the time the re-written gene doesn't express like wild-type, meaning that it is little more than a flip of the coin whether any given re-written gene will "work". I don't know that I would call this mastery of the rules of gene re-writing and this language should be toned down.

We have rewritten the essential and high-fitness genes of *Caulobacter crescentus*. Based on the limited set of constraints in the algorithm and the extent of rewriting, which includes synonymous recoding to up to 90% of triplets in a coding sequence, we would expect to have introduced mutations that alter transcription to a large extent. However, a prior expectation is difficult at best to formulate and substantiate, as rewriting and subsequent measurement have not been done at this scale. In that respect, we acknowledge that the reader might want to consider these number from the other perspective. In addition, we agree that the term "mastery" would raise the false expectation that we had prior knowledge on which genes would "work" at the level of transcription. We omitted the word "master" throughout the text and have altered the **Discussion** to better reflect this message:

The resultant similarities in both gene expression and transcription profiles tell us that our starting point, to a reasonable extent, allows for the reliable programming of a biological system.

Reviewer #2 (Remarks to the Author):

The manuscript describes the analysis of transcriptomic data derived from synthetic gene sequences, recoded to only preserve known regulatory motifs, in a natural host. As non-annotated regulatory sequence was open to recoding in the synthetic gene design, the authors hypothesised that altered expression of synthetic genes could be used to uncover novel regulatory mechanisms. By comparing transcript profiles of synthetic genes and natural genes in a *Caulobacter crescentus* host, the authors uncover previously cryptic sequence features affecting gene transcript levels. The authors also identify some possible novel post-transcriptional regulatory features. The approach is a powerful way to uncover the sequence-based mechanisms affecting gene expression and to help define design constraints for future synthetic gene and genome projects. The precise design constraints being proposed could be more clearly stated and would be better supported by more experimental evidence testing them.

We would like to thank the reviewer for the thoroughness of the review and the clear and useful comments. It has helped us to realize that we needed to take a next step in the formulation of design principles for synthesized DNA. We envision the reviewer will appreciate the extensive revision we have undertaken on two fronts:

- We rewrote large portions of the manuscript for clarity and to be able to capture a broad audience. Amongst others, we now formulate changes and rules as "Designer Changes" (**Figure 1 a**) and "Design Principles" (**Figure 2**).
- Based on the design principles we suggest, we redesigned, synthesized and assembled a subset of 82 redesigned genes. The segments were conjugated into *C. crescentus*, and the strains were subjected to RNA-Seq. Analysis of transcription level and coverage shape allowed us to confirm several of the hypotheses mentioned in the manuscript (**Results**

section “Implementation of the genome design principles improves the transcriptional landscape of synthesized DNA”).

Comments

Several statements are made about this work enabling “iteration-free programming of biological systems with synthesized information” and that the authors “master the rules of robust programming of a biological system to a great extent” and similar. This is a major claim and I do not think that the manuscript demonstrates this at all. Whilst the methodologies to screen gene designs for transcriptional and translational perturbations show interesting trends and individual cases, there isn't a great deal of direct experimental evidence to back these hypotheses as complete general rules for sequence design. There is still a large gap between the findings of this manuscript and some of the claimed implications. As such, they should be substantially toned down.

We have come to realize that the term “mastery” wrongfully implies that we would know how to build a synthetic genome from scratch in a following attempt. We did not have this intention. We meant to imply that the research community has gathered information over decades that enabled us to set a limited amount of constraints and implement a massive amount of nucleotide changes without destroying gene expression. We also meant to imply that we have learnt how to better build a next version of this genome, with lessons that will bridge to other organisms, as regulatory elements in these genomes follow the same logic and to a certain extent employ similar sequence motifs (the hexanucleotide -35 sequence and -10 sequences, Rho-independent transcription termination elements and Shine Dalgarno sequences). As a result of these considerations, we toned down the statements accordingly, for instance in the **Discussion**:

The resultant similarities in both gene expression and transcription profiles tell us that our starting point, to a reasonable extent, allows for the reliable programming of a biological system.

In addition, we have adapted the text to better reflect the path forward from what these findings enable and how subsequent work might bring us to the final goal - iteration-free programming of biological systems with synthesized information, for instance in the **Discussion**:

The design principles that we contribute here give genome engineers a higher likelihood of success when rewriting bacterial genomes. (...) Control elements and RNA degradation together govern gene expression. As such, we contribute to a map of the nucleotide sequence-based information space. In due time, this map will allow to program biological systems with synthesized information, allowing to explore possibilities that evolution has not yet touched upon.

The authors find several relationship trends between certain types of sequence and effects on transcription, and from there imply that resulting design constraints can be applied to future synthetic gene and genome design. The manuscript could be clearer on exactly what these constraints are and how they could be applied to the design process. Perhaps a table listing the design constraints emerging from this work would be useful to clarify this.

We included **Figure 2** and **SI Table S4**. The constraints that emerge from this work contribute to the design of *C. eth* 3.0.

What proportion of the “faulty” genes have sequence changes that can be specifically identified by any new design constraints as potentially causing a problem with transcription and translation? Conversely do any of the constraints, if retroactively applied to the synthetic genes, predict that genes should show disruption of function when they do not?

The constraints are a precautionary measure rather than a guarantee that a gene will be “functional”. In addition, the design constraints are aimed not only to create a genome with functional genes, but work on a higher level: we aim to create a robust genome with functional genes. As such, design constraints that prevent aberrant transcription from emergent transcription start sites in a synthetic genome might not contribute to gene function, but do pose control over gene expression. We revised the wording and we call the constraints “Design Principles” to reflect that we see this as a starting point. In addition, we test 3 of 5 principles and can confirm our initial reports with respect to omitted

regulatory elements, transcription termination elements, introduced promoter elements and polycistronic coupled genes, on which we report in **Results section “Implementation of the genome design principles improves the transcriptional landscape of synthesized DNA”**. In subsequent experiments that fall out of the scope of this manuscript, we envision to test a complete redesign at the translational level and the phenotypic level.

The manuscript discusses the potential causes of changes in transcript coverage shape in the “Sequence-based information in native CDS is captured in the major fraction of recoded CDS” section, identifying common sequence features that may be causing these changes. It is claimed “we can pinpoint which mechanisms underlie changes based on sequence...” but I cannot see that this is substantiated by any strong experimental evidence. To make these claims, the authors should make alterations to the synthetic genes to comply with their new constraints to show that they can alleviate detrimental transcriptional changes.

We realize that the phrasing is an inaccurate reflection of which mechanisms have and have not been experimentally validated. The statement was removed upon revision. Overall, we substantially tuned down such claims.

In addition, we performed additional experiments in which we show for 4 measured genes to indeed have excluded termination elements in the rewritten genome. We included a new **Results section “Implementation of the genome design principles improves the transcriptional landscape of synthesized DNA”** and adapt **SI Table S2**, in which we highlight these findings:

Transcription termination of 4 genes for which we predicted to have omitted Rho-dependent or -independent termination elements was probed by the addition of native termination elements or by repositioning the genes involved (SI Table S2). Comparison of the transcription curves of rewritten and redesigned genes confirmed our annotation of transcription termination elements (SI Table S2).

Table 1 shows that of the genes classed as functional, ~43% show differences in coverage shape. Of the genes classed as faulty, ~48% show differences in coverage shape. Can this really be said to support coverage shape as a particularly powerful way to identify changes that affect functionality? Why is the frequency of difference in coverage shape so similar for functional genes?

Transcription coverage shape and functionality measurements are complementary measurements that allow to understand discrete aspects of the rewritten genome. Functionality measurements allow for a binary classification: a category of genes work at the plasmid level and a category of genes does not work. Transcription coverage shape is the only high-throughput method that allows to notice sequence-based changes in the coding sequence. In addition, it allows to distinguish genes for which regulation occurs at the post-transcriptional level: “faulty” genes with no reported issues at the transcription level. Finally, we improve the robustness of genome design with the transcriptional landscape. As such, “particularly powerful” would not be our choice of words, but we did not come across any other method that would allow to identify fundamental and engineered regulatory elements in transcription and translation for an entire genome.

In the manuscript, you say that faulty genes with the same coverage shape must have differences in translation causing the defect. One of these genes is investigated in depth, but did you find any indications in the other sequences as to what could be causing the problems?

We performed computational analysis (structural analysis, recognition of SD motifs and conservation analysis) that strengthen our hypothesis with respect to the importance of structural elements at the RNA level in 19 additional genes.

The manuscript is quite difficult to read in places. I’d recommend a further proof reading. There are a few sentences like “Orthogonal coverage shape comparison and functionality measurements of the recoded bacterial genome of C. eth 2.0 show to uncover novel regulatory features, RNA-based control elements” that would benefit from reworking.

We performed an extensive revision of the manuscript.

The terminology around synthetic genomes and *C. eth 2.0* isn't always accurately employed in this manuscript and I think that this could be confusing for the reader. The work in this manuscript takes a handful of synthetic genes at a time and characterises them in the context of a natural host cell. Even if you accept that a DNA sequence that doesn't allow for cellular growth constitutes an existing synthetic organism, as claimed in the introduction, the characterisation those genes in small batches in a natural host context is hard pushed to be called a complete *C. eth 2.0* transcriptome. The authors should be careful not to imply that a collection of genes separately characterised constitute a synthetic organism that has been characterised.

We agree that the batch measurement of a genome both at the genetic and transcription level pose certain drawbacks. We meant to imply that we measured the gene expression of all genes of *C. eth 2.0* and adapted the **Introduction** accordingly:

Collectively, we compared 612 protein-coding and non-coding genes by RNA-Seq (Nagalakshmi et al., 2008).

We included a statement in the **Discussion** on the imposed drawbacks of batch measurements, see the response to the comment "The synthetic genes are tested (..) any such effects?" below.

Line 317-318 – "These guidelines are relevant for any bacterial genome" – A fairly important point is only very briefly addressed here. *C. crescentus* is undoubtedly a good organism to perform this study in, given the tight control of gene copy number, but a little more discussion on how applicable these findings might be for other organisms would be good. Why can we assume that the guidelines derived are applicable to any bacterial genome? This is obviously a major factor in the wider impact of this study.

We are excited to hear that the reviewer recognizes the potential value of this work for other organisms and are happy to elaborate. In the **Discussion**, we now include the specific lessons that we expect bridge to other organisms and the reasoning we employ to back such expectations (**Line 431-440**).

The synthetic genes are tested in sections rather than individually. Is it possible that being co-located on a DNA section with a particular transcription factor could have an effect on a gene's expression? In such a situation, there would be increased levels of the transcription factor, leading potentially to changes in the background gene expression profile of the cell or possibly even direct effects on the expression of a gene being analysed. Is there any indication in your data of any such effects?

For a thorough analysis on the effect of (1) the addition of DNA and (2) the addition of gene copies to the cell, biological replicates should be taken. We conducted an initial analysis in which we probed for copy number-dependent up- or downregulation of native genes. The main effect we observed was related to (1), in that sense that metabolic pipelines responsible for nucleotide synthesis appeared to be upregulated. For (2), we indeed hypothesized that the presence of a second copy of a gene might affect the native gene or perhaps other native interactors. With the limited replicates at hand for (2), we cannot substantiate such an effect. We consider the experimental setup that would allow to point out such interactions to be out of the scope of this work, however, we value the comment will take it into consideration in a future setup.

Line 191 – You state that you exclude 10 genes from further analysis due to mutations introduced in synthesis or assembly. Why weren't they fixed to allow their inclusion?

We did not fix these genes mainly for two practical reasons: the mutations had not been observed at the time of RNA-Seq measurements. In addition, the genes are not sequential. Fixing them would require synthesis, assembly, generation and measurement of up to 10 strains of merosynthetic *C. crescentus*. The beauty of it is in that precisely these mutations show that coverage shape and functionality measurements complement one another: these genes would indeed not show problems in transcription, but in translation, which is precisely the category that they were mapped in before realizing that mutations had occurred.

Construction of merosynthetic *Caulobacter crescentus* - 7 of the 37 parts were tested in tandem with another part. Segment 9 was used as a control as it was present in two strains in combination with 2

other different segments. Why didn't the authors include segment 9 alone as a control strain? Wouldn't this be a good way to determine any effects of having 2, rather than 1, segment in the cell? It might also give an indication as to whether other synthetic genes co-expressed in the same cell have an effect on other synthetic gene expression profiles.

First, we performed a control that showed that the NGS of the mRNA derived from SDRNA does not differ from the mRNA derived from commercially available methods. We used segment 9,10. Second, we probed whether the experimental and computational pipeline delivered reproducible results. Here, we indeed used segment 9 in two different contexts (segment 8,9 and segment 9,10). Finally, we measured segment 34 as a single segment and in combination as segment 33,34 to probe if the pooled approach we used at a later stage of the experimental process would allow to assign reads to their respective segments. This pooled approach differed from the initial approach in that samples were either pooled at harvest or after RNA extraction. In the initial approach, samples were not pooled.

Please be referred to the response to the remark "The synthetic genes are tested (..) any such effects?" on the effect of the addition of DNA and the addition of gene copies to the cell.

Sample preparation for RNA-Seq - "Merosynthetic *C. crescentus* were harvested at an OD600 of 0.4 or at mid-log phase" is quite vague. Does this indicate that samples were consistently harvested at OD600 0.4, which is mid exponential, or were different samples harvested using different criteria? Several other parts in this section are not as clearly explained as they could be, e.g the description of the "SKYLINE" method and the process described in lines 394-396. This section is the describing the acquisition of the key data underpinning this study and so it's important that the methodology is clear.

The aim was to harvest samples in the exponential phase and preferably at and OD600 of 0.4. SI Dataset 1 contains all measured values at the time of harvest. We adapted **Methods section "Sample preparation for RNA-Seq"** accordingly:

Merosynthetic C. crescentus were harvested in the exponential phase (..).

We now elaborate on the methodologies that we have employed in **Methods section "Computational analysis of data acquired through RNA-Seq"**, e.g. in the new subsection **"Acquisition and analysis of transcription curves"**.

Gene exclusion principles – I'm confused as to what the numbers in brackets represent. Are these gene numbers mutually exclusive or are they additive? It states elsewhere that 506 genes were included in the analysis but it's not clear how many in total were excluded.

We sequentially excluded genes from the analysis based on predefined exclusion principles. We acknowledge that the sequential exclusion, with the mentioning of the remaining genes, is confusing. We adapted the **Methods section "Gene exclusion principles"** accordingly.

Construction of strains for the validation of post-transcriptional regulation in CCNA_00197 – I may have missed it, but I can't see the sequences of the gBlocks used or the constructs generated in the supplementary information. These sequences should be included and their location referenced in this methods section.

We included a supplementary file **gBlock Table.xlsx** that contains this information.

β -galactosidase reporter assay – This section states that the error bars have been omitted in Figure 4b and 4c (and I'm presuming also in Figure S4). A version of these figures with error bars should also be included in the supplementary information for reference.

We included error bars in both **Figure 5 b** (was: 4 b and c) and **SI Figure S7** (was: SI Figure S4), where we had mentioned to omit error bars for clarity.

Figure 1 – The layout of this figure is quite confusing. The *Caulobacter ethensis* genome is not obviously part of subfigure 1c and it took me a while to notice that the colouring of the gene markings

on the outer rings corresponded to the fold change in expression. It might be clearer to make the genome map its own subfigure with an additional fold change scale bar to reiterate this.

We repositioned the subpanels in **Figure 1** and added an in-figure legend to panel d to point the reader to the color and fold change relation.

Figure 2a – I appreciate that the shape of coverage is the important thing being compared here, but the different scaling presumably being applied to the synthetic and natural data could be a bit misleading. I'd recommend either adding some sort of scale to the y-axis or adding a third plot for each gene showing an overlay of the two curves to clarify exactly what's being shown.

The y-axis scale has been added to **Figure 3 a** (was: 2 a) to indicate how gene expression compares between the native and rewritten gene.

Figure 2b – It's not clear, either here or in the corresponding manuscript text, where these bases actually lie in the CDSs. It would be useful to indicate the positions of the bases.

We now indicate the precise position of the consensus motif in the CDS in **Figure 3 a**.

Figure 2c – In the text, this is referenced in the sentence "Detailed examination of the sequence of *ftsW* (CCNA_02635), where the coverage shape shows differences in the coding region, reveals the interplay of such processes (Fig. 2 c)". So far as I can see, *ftsW* does not feature in this figure at all.

We apologize to have overlooked the admission of the figure in revision. **SI Figure S5 b** features *ftsW*.

Figure 2d – This shows a proposed mechanism that I don't think is mentioned at all in the manuscript text. If a mechanism is being proposed, it should be discussed in the text.

The mechanism has been suggested in previous studies (Dühring *et al.*, 2006, Lasa *et al.*, 2011) and fits our experimental observations. We adapted the **Results section "Transcriptional interference and synonymous codon changes affect RNA stability"** to better accommodate the studies, the figure, and the fact that these are observations that would fit previous studies, yet have not been validated:

(..) cis-encoded RNA, or antisense RNA originating from a promoter on the opposite strand is thought to be able to interfere with sense transcription through the creation of double-stranded substrates that may form a target for endonucleases RNase III and E (Dühring et al., 2006, Lasa et al., 2011).

Figure 4a – The expanded sequence showing the stop/start junction of the two CDSs is displayed in a way that readers might initially take to be the two DNA strands. I'd recommend tweaking the style of this.

We adapted **Figure 5 a** (was: Figure 4) to better reflect the same-strand start-stop structure.

Figure 4b/c – The splitting of these two sub-figures is confusing, I'd recommend just making the whole thing one sub-figure.

We joined the subpanels into a single panel in **Figure 5 b** (was: Figure 4 b and c).

Reviewers' Comments:

Reviewer #1:

Remarks to the Author:

Re-review

The authors have simplified the language and revised the manuscript substantially making the text more accessible. The new data in section "Implementation of the genome design principles improves the transcriptional landscape of synthesized DNA" addresses the major critique of the initial review, which was to provide data testing the hypotheses associated with the genome rewrite. Unfortunately, the data associated with this section are largely buried in the supplemental information and are poorly described in the main text. The authors should revise this section and present a more rigorous description in the main text along with a figure, as this is the most important part of their story. This change should be made before publication.

Below are additional comments to be addressed before publication.

1. There is a peculiar new emphasis on genetic code expansion via codon minimization throughout the introduction. While interesting, it isn't particularly relevant to the topic of this manuscript, which is the functional annotation of regulatory elements using a synthetic genomics approach. The authors could consider a much more concise introduction to the topic at hand.
2. Line 20-23 – while the authors optimized their design to minimize DNA synthesis costs, this is not everyone's goal is genome rewriting projects. The wording in these two sentences should be clarified as it is too broad.
3. Line 27 – not clear what the "virtual space" refers to.
4. Line 33 – clarification needed - recalcitrant to what?
5. Line 41-55 – not obvious where results from the previous publication end and new results in this manuscript begin – provide clarification for reader. I had to go back to the previous publication to remind myself of the design strategy – I recommend including a summary of that strategy here so others don't have to search out the details.
6. Line 44 – biological blocks isn't meaningful and needs a definition.
7. Line 49 – would be interesting to indicate what percent of all codons 123,563 represents.
8. Line 82-82 –4.21% seems high? If each plasmid encodes 20 genes and the wild-type genome encodes ~3700, shouldn't this be more like 0.5%? what accounts for the 10x boost.
9. Line 85 – change to "3 classes of Designer changes".
10. Line 180-181 – sentence not clear.
11. Line 195 – not clear what "prompt" means here.
12. Line 200 – should be "designed sequences".
13. Line 204 – should be "can erase" instead of "erases".
14. Line 222 - Table 1 is not readily understood and requires additional text either in the manuscript or as a footer to the table.
15. Line 373-414 – where is the data to support this section? What about Design Principles 4 and 5?
16. Line 383 – define merosynthetic or reword.
17. Line 415 – sentence needs editing – essential proteins instead of amino acid?

Reviewer #2:

Remarks to the Author:

The manuscript has clearly undergone a substantial rewrite and its content, clarity and claims have been much improved. I'm particularly pleased to see the additional experimental validation of the rules derived from the transcriptional analysis of the merosynthetic constructs. I think that this really strengthens the work. All of my previous concerns have been addressed by the revisions or the authors' rebuttal. I think that the manuscript shows a really interesting approach to using merosynthetic DNA to both uncover an organism's existing cryptic regulatory elements and to

hone design principles for future synthetic constructs and genomes. As, such I recommend the revised manuscript for publication.

REVIEWERS' COMMENTS

Remarks to the Reviewers from the Authors:

We would like to thank Reviewer #1 and Benjamin Blount for the time and effort spent in the critical assessment of the manuscript, both in the first and second revision. Particularly in the first stage, the clear and thorough feedback initiated a complete revision of the manuscript. We would like to emphasize that we very much appreciate their role in this process.

Reviewer #1 (Remarks to the Author):

Re-review

The authors have simplified the language and revised the manuscript substantially making the text more accessible. The new data in section "Implementation of the genome design principles improves the transcriptional landscape of synthesized DNA" addresses the major critique of the initial review, which was to provide data testing the hypotheses associated with the genome rewrite. Unfortunately, the data associated with this section are largely buried in the supplemental information and are poorly described in the main text. The authors should revise this section and present a more rigorous description in the main text along with a figure, as this is the most important part of their story. This change should be made before publication.

In light of the concerns raised by Reviewer #1 and the appreciation for the section in the re-review of Reviewer #2, we revised **Results section "Implementation of the genome design principles improves the transcriptional landscape of synthesized DNA"** to describe the results obtained in more depth. In addition, we revised **Figure 2** to depict results of our measurements of the redesigned segments.

Below are additional comments to be addressed before publication.

1. There is a peculiar new emphasis on genetic code expansion via codon minimization throughout the introduction. While interesting, it isn't particularly relevant to the topic of this manuscript, which is the functional annotation of regulatory elements using a synthetic genomics approach. The authors could consider a much more concise introduction to the topic at hand.

The **Introduction** has been revised to better reflect the topic of the manuscript.

2. Line 20-23 – while the authors optimized their design to minimize DNA synthesis costs, this is not everyone's goal is genome rewriting projects. The wording in these two sentences should be clarified as it is too broad.

The wording was adapted to better reflect the communal goals as follows. "We can program biological systems with DNA that is based on native nucleotide sequences. To enable DNA synthesis and assembly, and to provide room for the creation of orthogonal genetic codes, native DNA can be rewritten."

3. Line 27 – not clear what the "virtual space" refers to.

We omitted the word "virtual". The word "virtual" was meant to ensure that the reader understands that the "information space" for sequence-encoded gene regulatory features is not physical.

4. Line 33 – clarification needed - recalcitrant to what?

The phrase was revised to "certain triplets prove to be recalcitrant to synonymous exchange."

5. Line 41-55 – not obvious where results from the previous publication end and new results in this manuscript begin – provide clarification for reader. I had to go back to the previous publication to remind myself of the design strategy – I recommend including a summary of that strategy here so others don't have to search out the details.

We adapted the paragraph to include a clear distinction of the present study with the phrase “For the present study, we reasoned ..”.

6. Line 44 – biological blocks isn't meaningful and needs a definition.

We included the following description of biological blocks in the **Introduction**: “.. defined chunks of DNA that contain one or several genes that belong together from a gene regulatory point of view.”

7. Line 49 – would be interesting to indicate what percent of all codons 123,563 represents.

Addition of “collectively recoding 56.1% of all codons.”.

8. Line 82-82 –4.21% seems high? If each plasmid encodes 20 genes and the wild-type genome encodes ~3700, shouldn't this be more like 0.5%? what accounts for the 10x boost.

The plasmid copy number and transcription from introduced transcription start sites (**SI Fig. S2 c**) account for the boost. We correct for the plasmid copy number at a later stage and before analysis of transcription (**SI Dataset 1 Sheet 2**).

9. Line 85 – change to “3 classes of Designer changes”.

The statement has been revised.

10. Line 180-181 – sentence not clear.

Revised to “Single point mutations may have considerable effects in a genome when sequences in that genome closely resemble promoter elements. In this situation, mutations allow for the emergence of novel gene products in amenable regions (Yona *et al.*, 2018).”.

11. Line 195 – not clear what “prompt” means here.

Replaced with “ample” to reflect the extensive amount of introduced promoters as a result of sequences that already closely resemble promoter elements in combination with the extensive amount of genome rewriting.

12. Line 200 – should be “designed sequences”.

The statement has been revised.

13. Line 204 – should be “can erase” instead of “erases”.

The statement has been revised.

14. Line 222 - Table 1 is not readily understood and requires additional text either in the manuscript or as a footer to the table.

A footer has been added to **Table 1** and reads as follows: “In previous work, we measured if a recoded, rewritten gene would be able to functionally replace the native gene or is faulty (Venetz *et al.*, 2019). The results for 471 genes are contained in the rows “Functional” and “Faulty”. The

columns refer to the counts for these same genes with respect to comparison of transcription curves between native and rewritten genes as observed after measurements described in this work.”

15. Line 373-414 – where is the data to support this section? What about Design Principles 4 and 5?

We revised **Results section “Implementation of the genome design principles improves the transcriptional landscape of synthesized DNA”** to describe the results obtained in more depth and added measurement results to **Figure 2**. We now omit Results section “Transcriptional interference and synonymous codon changes affect RNA stability”, corresponding to Design Principle 4, to adhere to journal formatting requirements, i.e. the maximum of 6’000 words. The section is integrated, in part, into Results section “Sequence rewriting inadvertently introduces promoter elements that lead to aberrant transcription” and is, in part, included in the Supplementary Information under Supplementary Notes. We discuss and validate Design Principle 5 (now: Design Principle 4) in Results section “Translation of large subunit ribosomal protein L19P, RplS is facilitated by a putative conserved mRNA hairpin in the coding region of tRNA methyltransferase TrmD”.

16. Line 383 – define merosynthetic or reword.

Rephrased to “We assembled the synthesized DNA into 4 additional segments, introduced these segments into *Caulobacter*, creating 4 merosynthetic strains that each carry a subset of genes in twofold, and measured transcription by RNA-Seq (Methods, SI Dataset 3).”

17. Line 415 – sentence needs editing – essential proteins instead of amino acid?

Line 417 has been adapted to read “.. we retained only annotated control elements, and the amino acid sequence of protein coding genes.”

Reviewer #2 (Remarks to the Author):

The manuscript has clearly undergone a substantial rewrite and its content, clarity and claims have been much improved. I’m particularly pleased to see the additional experimental validation of the rules derived from the transcriptional analysis of the merosynthetic constructs. I think that this really strengthens the work. All of my previous concerns have been addressed by the revisions or the authors’ rebuttal. I think that the manuscript shows a really interesting approach to using merosynthetic DNA to both uncover an organism’s existing cryptic regulatory elements and to hone design principles for future synthetic constructs and genomes. As, such I recommend the revised manuscript for publication.

Benjamin Blount